# Control of intracellular pH and bicarbonate by $CO_2$ diffusion into human sperm

Elena Grahn[1,7], Svenja V. Kaufmann[2,7], Malika Askarova[1], Momchil Ninov[2,3], Luisa M. Welp[2,3], Thomas K. Berger ®[1,4] ✉, Henning Urlaub ®[2,3,5] ✉ & U.Benjamin Kaupp ®[1,6] ✉

The reaction of $CO_2$ with $H_2O$ to form bicarbonate ($HCO_3^-$) and $H^+$ controls sperm motility and fertilization via $HCO_3^-$-stimulated cAMP synthesis. A complex network of signaling proteins participates in this reaction. Here, we identify key players that regulate intracellular pH ($pH_i$) and $HCO_3^-$ in human sperm by quantitative mass spectrometry (MS) and kinetic patch-clamp fluorometry. The resting $pH_i$ is set by amiloride-sensitive $Na^+/H^+$ exchange. The sperm-specific putative $Na^+/H^+$ exchanger SLC9C1, unlike its sea urchin homologue, is not gated by voltage or cAMP. Transporters and channels implied in $HCO_3^-$ transport are not detected, and may be present at copy numbers < 10 molecules/sperm cell. Instead, $HCO_3^-$ is produced by diffusion of $CO_2$ into cells and readjustment of the $CO_2/HCO_3^-/H^+$ equilibrium. The proton channel $H_v1$ may serve as a unidirectional valve that blunts the acidification ensuing from $HCO_3^-$ synthesis. This work provides a new framework for the study of male infertility.

Fertilization is a complex multi-stage process. After ejaculation, a maturation process, called capacitation, sets in, whereby sperm become motile and acquire the ability to locate and fertilize the egg. During capacitation, sperm adopt a whiplash-like flagellar beat pattern, called hyperactivation, that is required to penetrate the protective egg investment[1–3]. Arguably, the most important cellular reaction that controls motility and capacitation of sperm is the $CO_2 + H_2O \rightleftarrows HCO_3^- + H^+$ equilibrium ($CO_2/HCO_3^-/H^+$). $HCO_3^-$ and $H^+$ feed into a sprawling network of signaling proteins that translate the state of this equilibrium into changes in cAMP and $Ca^{2+}$, the intracellular messengers (for reviews, see ref. [4–8]). The $CO_2/HCO_3^-/H^+$ reaction is catalyzed by carbonic anhydrases (CA). The extra- and intracellular $CO_2/HCO_3^-/H^+$ equilibria are coupled across membranes not only in sperm but also surrounding epithelia cells by $CO_2$ diffusion, and $HCO_3^-$ and $H^+$ transport via ion channels and solute carriers (SLC).

Three key events reportedly are required for capacitation: first, an increase of intracellular $HCO_3^-$ concentration that stimulates cAMP synthesis by a $HCO_3^-$-activated soluble adenylate cyclase (sAC)[9], which initiates protein kinase A (PKA)-mediated phosphorylation that, among other events, enhances flagellar beat frequency[2,5,10,11]. A second hallmark of capacitation is an increase of intracellular pH ($pH_i$), which activates the exquisitely pH-sensitive, sperm-specific $Ca^{2+}$ channel CatSper[12]. The $Ca^{2+}$ influx is important for sperm hyperactivation[13], although the extent of this regulation varies among mammals. Moreover, $Ca^{2+}$ and alkaline $pH_i$ activate the sperm-specific Slo3 $K^+$ channel in human and mouse, respectively[14,15]. Finally, hyperpolarization of the membrane potential reportedly is key for capacitation[2]. The

[1]Max Planck Institute for Neurobiology of Behavior—caesar, Molecular Sensory Systems, Ludwig-Erhard-Allee 2, 53175 Bonn, Germany. [2]Max Planck Institute for Multidisciplinary Sciences, Bioanalytical Mass Spectrometry, Am Fassberg 11, 37077 Göttingen, Germany. [3]University Medical Center Göttingen, Institute of Clinical Chemistry, Bioanalytics, Robert-Koch-Strasse 40, 37075 Göttingen, Germany. [4]Department of Neurophysiology, Institute of Physiology and Pathophysiology, Philipps-University Marburg, Deutschhausstrasse 1-2, 35037 Marburg, Germany. [5]Cluster of Excellence, Multiscale Bioimaging: from Molecular Machines to Networks of Excitable Cells (MBExC), University of Göttingen, Göttingen, Germany. [6]Life & Medical Sciences Institute (LIMES), University Bonn, Carl-Troll-Strasse 31, 53115 Bonn, Germany. [7]These authors contributed equally: Elena Grahn, Svenja V. Kaufmann. ✉e-mail: thomas.berger@uni-marburg.de; henning.urlaub@mpinat.mpg.de; ubkaupp@uni-bonn.de

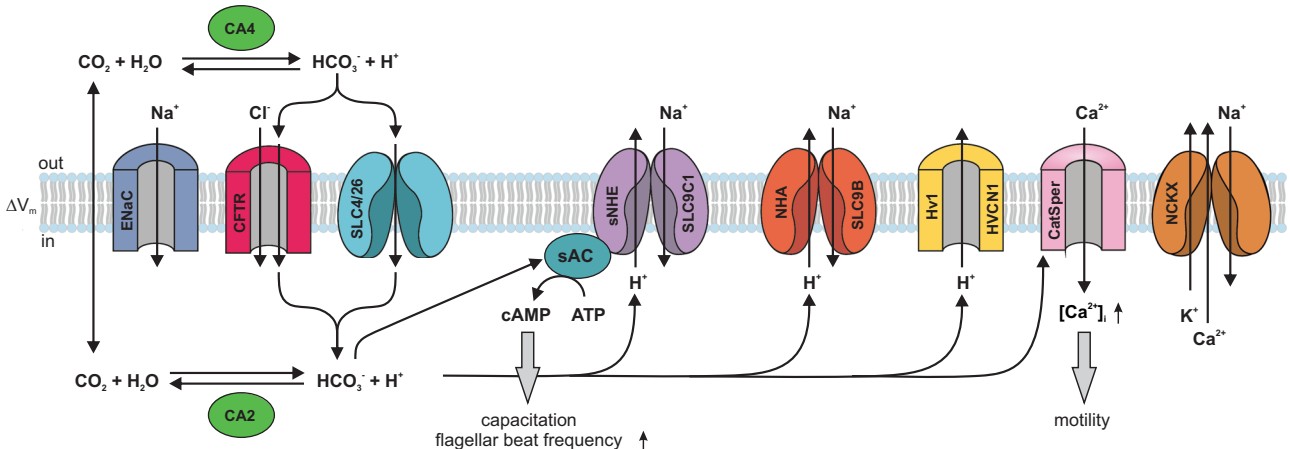

**Fig. 1 | Putative molecules and mechanisms of $pH_i$ and $HCO_3^-$ regulation in mammalian sperm.** ENaC epithelial $Na^+$ channel, CFTR cystic fibrosis transmembrane regulator, SLC4 and SLC26 solute carrier families 4 and 26 that transport $HCO_3^-$ and anions; CA carbonic anhydrase 2 and 4, sNHE sperm-specific sodium/ hydrogen exchanger SLC9C1, sAC soluble adenylate cyclase, NHA sodium/hydrogen antiporters of the SLC9B family, $H_v1$ proton-selective ion channel, CatSper sperm-specific cation channel. NCKX, $Na^+$- $K^+/Ca^{2+}$ exchanger. The figure was designed by Dr. René Pascal.

physiological consequences of the hyperpolarization are not precisely known.

Many proteins and mechanisms have been implied in the execution and coordination of these three signaling events (Fig. 1)[2,4]. $HCO_3^-$ has been proposed to enter sperm via transporters of the solute carrier families SLC4 (ref. [16,17]) and SLC26 (ref. [18–22]). $HCO_3^-$ could also enter through a $Cl^-$ channel, the cystic fibrosis transmembrane regulator (CFTR)[18,23–27]. Alternatively, CFTR might balance $Cl^-$ fluxes associated with $Cl^-/HCO_3^-$ exchange, and reportedly affects $HCO_3^-$ transport by interactions with SLC4 and SLC26 transporters[28] and with the epithelial sodium channel (ENaC)[27]. Finally, $HCO_3^-$ could be synthesized from $CO_2$ that enters by diffusion across the cell membrane. The mechanism by which $HCO_3^-$ rises is crucial because it determines whether sperm alkalize or acidify.

The increase of $pH_i$ could be accomplished by several different mechanisms: $H^+$ efflux via the voltage-gated proton channel $H_v1$ (HVCN1)[29,30] or two different types of sodium/proton ($Na^+/H^+$) exchangers—sperm-specific sNHE (SLC9C1)[31–33] and the $Na^+/H^+$ antiporters NHA (SLC9B1 and SLC9B2)[34,35]. Alternatively, by feeding into and readjusting the $CO_2/HCO_3^-/H^+$ equilibrium, which is catalyzed by carbonic anhydrases CA2 and CA4 (ref. [36,37]), $HCO_3^-$ influx could alkalize sperm indirectly. Finally, hyperpolarization could be brought about by activation of Slo3 $K^+$ channels, $HCO_3^-$ or $Cl^-$ influx via CFTR channels, blockage of $Na^+$ inward currents via ENaC channels, or $H^+$ efflux via $H_v1$ channels. Because membrane potential $V_m$ is set by the balance of inward and outward currents, any one of these various channels may participate in producing the hyperpolarization (for review, see ref. [38]) In summary, the literature abounds with competing hypotheses how the increase of $HCO_3^-$, alkalization, and hyperpolarization is generated. This is not surprising, considering that many of the chemical reactions and molecular components are coupled and cannot be varied independently from each other.

The complexity is all the more perplexing as both the properties of these signaling molecules and the protein inventory of sperm differ markedly among mammalian species[6]. $H_v1$ channels exist in human and boar but not mouse sperm[30,39]. The $K^+$ channel Slo3 is controlled by $pH_i$ in mouse[15,40,41] but primarily by $Ca^{2+}$ in humans[14]. The $Ca^{2+}$ channel CatSper is activated by the female sex hormone progesterone in human[42,43] but not mouse[42]. 

Considering the diversity of sperm-specific molecules and the uncertainty of their actions, it is a daunting task to unravel this perplexing cellular $pH_i/HCO_3^-$ network, and the generalization of concepts across phyla must be viewed with due caution[6].

Our present knowledge about the presence of these signaling proteins in human sperm is largely based on immunodetection, the pharmacology of signaling components, and on concepts derived from knockout-mouse lines or other mammalian species. However, Fc binding sites on sperm can confound immuno-detection, and some drugs display multiple off-target actions[44–47]. Moreover, fertilization phenotypes may result from defects in spermiogenesis, and targeted proteins that exist in precursor cells may be absent from mature sperm.

Here we examine the presence or absence of signaling proteins in human sperm using quantitative liquid chromatography (LC)-coupled mass spectrometry (MS). We determine the copy number of several signaling proteins and, using limit-of-detection (LOD) values, we estimate the maximal copy number for those proteins, which we could not detect. The large dynamic range between copy numbers and LOD values seriously constrains hypotheses about the physiological function of molecules that are either absent or of low abundance. We corroborate the MS data by functional measurements of $H^+$ and $Na^+$ fluxes in single sperm cells using fluorescence microscopy and patch-clamp fluorometry. We show that $CO_2/HCO_3^-$ acidifies rather than alkalizes sperm, and that synthesis of $HCO_3^-$ from $CO_2$ dominates $HCO_3^-$ entry via $HCO_3^-$ solute carriers. Finally, we propose that $H^+$ efflux by the $H_v1$ proton channel keeps the acidification resulting from $HCO_3^-$ synthesis at bay and, thereby, sustains high $HCO_3^-$ levels inside sperm.

## Results and discussion
### Identification of signaling molecules by mass spectrometry
Several independent proteomes have been assembled from human sperm[48–51] that differ markedly in size and in key signaling components (Supplementary Table 1) (see summary in[50]). In addition, targeted MS studies of human sperm have also been performed[52]. Here, we determined the proteome from three different donors by LC-MS/MS (Supplementary Data 1). Only swim-up sperm were used to avoid contamination by somatic cells and non-motile precursor sperm cells. In three sperm samples, we detected ~5000 proteins on average, similar to the[51] proteome but ~4–5-fold more proteins than in the other studies (Supplementary Table 1). A similarly deep proteome has been recently reported for *Drosophila* sperm[53,54]. Of all identified proteins in our study, 63% were detected in all three sperm samples (Fig. 2a). Wang et al. (2013) reported an overlap of 82% of proteins identified in their analysis; of note, each of their replicates consists of sperm samples mixed from several donors, whereas we have analyzed proteomes from single donors.

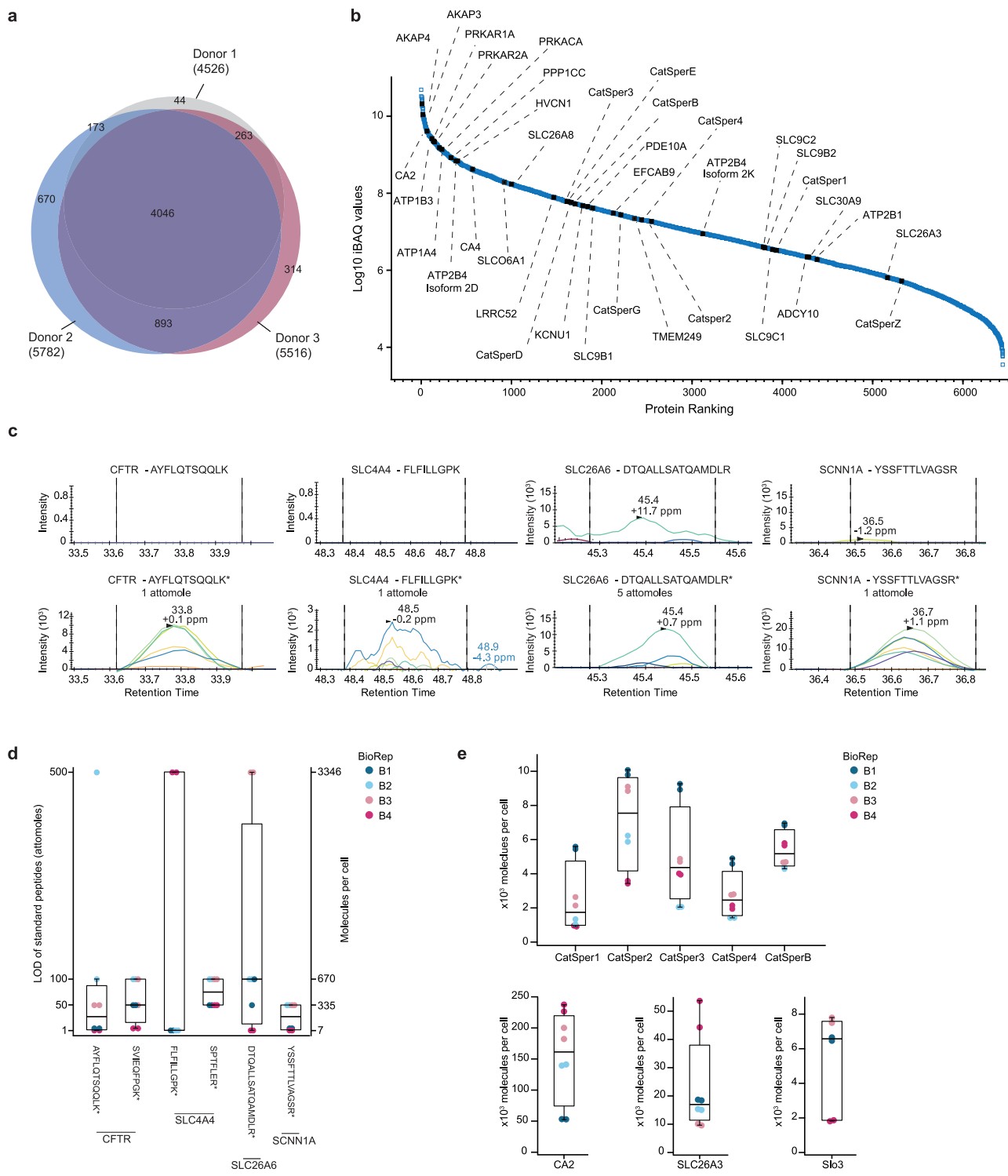

We searched our data notably for those proteins that had been implied in transport of $HCO_3^-$, $Na^+$, $H^+$, $Cl^-$, $Ca^{2+}$, and in signaling of cAMP or $Ca^{2+}$ (Fig. 1). All signaling proteins were detected that had previously been identified in mammalian sperm by rigorous functional testing, including electrophysiological recordings, or by gene targeting in mice. Most of these proteins are considered sperm-specific and are absent from somatic cells. These proteins were assembled in a 'positive' list entailing 39 proteins (Supplementary Table 2). Another 'positive' list comprises 22 proteins that are

unrelated to the signaling scheme in Fig. 1 but which serve obvious sperm-specific functions such as acrosomal exocytosis, gamete recognition, or axonemal structure (Supplementary Table 3). Without exception, all sperm-specific proteins were detected, among them all nine CatSper subunits including the $Ca^{2+}$-binding protein EFCAB9 that interacts with CatSperZ[55] and associated proteins SLCO6A1, the human homolog of mouse SLCO6C1, and TMEM249 (ref. 56); the Slo3 channel and its auxiliary subunit LRRC52[14]; the $Na^+/H^+$ exchangers SLC9B1 and 2 (ref. 34), carbonic

**Fig. 2 | MS of human sperm and parallel reaction monitoring (PRM) MS analysis of standard peptides for determination of limit-of-detection (LOD) and quantification. a** Venn diagram of identified proteins (*n* = 3 biological replicates) by LC-MS/MS. **b** Ranking of proteins according to the abundance (*n* = 3) (iBAQ). For abbreviations of proteins see Supplementary Tables 2–4. **c** Transition channels of the most sensitive PRM analysis of peptides corresponding to SCNN1A, CFTR, SLC4A4 and SLC26A6. Peptide sequences are given; asterisks denote the [13]C- and [15]N-labelled amino acids in standard peptides. Upper panels: No specific PRM signals detected for endogenous peptides derived from 90,000 sperm cells; lower panels: PRM signals detected from 1 attomole (5 attomoles for SLC26A6) of standard peptide spiked into endogenous peptides derived from 90,000 sperm cells. Different colors of the transitions signify different fragment ions. **d** Box plot of LOD values of standard peptides representing proteins CFTR, SLC4A4, SLC26A6, and SCNN1A monitored in *n* = 4 biological replicates (BioRep B1–B4) of digested sperm sample. Sequences of the standard peptides are listed. Left y-axis shows detectable amount of standard peptide in attomole spiked into endogenous peptides derived from 90,000 sperm cells. Right y-axis shows the calculated molecules/cell corresponding to the amount of detectable standard peptides. **e** Box plot of protein molecules/cell as determined by PRM MS for carbonic anhydrase II (CA2), chloride/anion exchanger (SLC26A3), potassium channel (Slo3), CatSper channel subunits (CatSper1, CatSper2, CatSper3, CatSper4, and CatSperB). Biological replicates are indicated by color, copy numbers were determined in *n* = 4 biologically independent replicates (BioRep B1–B4) of digested sperm sample from four independent experiments. For some proteins the differences between replicates were smaller than the size of the data points, in those cases less than eight data points are visible. For Slo3, only data from three of the four biological replicates could be used. Box plots represent 25%, 50% (median), and 75% quartiles, with whiskers displaying minimum and maximum data points, except for outliers outside of 1.5-times interquartile range (IQR). Source data is provided as a source data file.

anhydrases CA2 and CA4 (refs. 36,37), and the H$_v$1 channel[30,39]. Regarding sNHE, we detected two isoforms—SLC9C1 and SLC9C2; SLC9C2 is an orphan protein that was discovered genomically but has not yet been detected or characterized in any cell type[57]. Most of these components are integral membrane proteins that are notoriously more difficult to detect than soluble proteins. Moreover, all enzymes and targets involving cAMP signaling were detected, including the soluble adenylyl cyclase sAC[9], the phosphodiesterase PDE10 (ref. 58), all catalytic and regulatory subunits of protein kinase A (PKA), the PKA-scaffolding proteins AKAP3 and 4, and the sperm-specific phosphatase PPP1CC[59].

The detection of seven CatSper subunits and three associated proteins in all three biological replicates (CatSper1 in two and CatSperZ in one replicate) speaks to the sensitivity, robustness, and reliability of our approach. In previous proteomic studies only CatSper1, 2, 4, B, and D were identified[50]. With very few exceptions, our MS data agree with a previously published proteome of similar size[51] (Supplementary Tables 2–4).

The most abundant components according to iBAQ values[60] are PKA subunits, the anchor proteins AKAP3 and 4, carbonic anhydrase CA2 and 4, the Na$^+$/K$^+$-ATPase subunits ATP1A4 and ATP1B3, the Ca$^{2+}$-transporting ATPase ATP2B4, the phosphatase PPP1CC, and the proton channel HVCN1 (Fig. 2b).

Unexpectedly, we could not detect many proteins implied by previous studies either directly or indirectly in HCO$_3^-$ transport, including CFTR, the ENaC sodium channel, and most members of the HCO$_3^-$- and anion transporter families SLC4A and SLC26, except SLC26A3, a chloride/anion exchanger, and SLC26A8, a testis-specific sulfate transporter preferentially expressed in precursor cells[19,61]. Furthermore, we were not able to detect Na$^+$/Ca$^{2+}$ and Na$^+$/Ca$^{2+}$-K$^+$ exchangers that had been invoked in Ca$^{2+}$ homeostasis, and the K$^+$ channel Slo1 (ref. 62) or its auxiliary ß and γ subunits. Proteins that had been commonly implied in HCO$_3^-$/pH$_i$/Cl$^-$ sperm signaling but were not detected by LC-MS/MS in all three samples are assembled in a 'negative' list (Supplementary Table 4). This list also includes many anion transporters that could act as potential HCO$_3^-$ transporters. Conversely, if a protein does not appear on the negative list, it does not imply that we detected that protein. Although we cannot exclude that one or another of the 37 proteins in the "negative" list was missed in our LC-MS/MS analysis, these proteins have been successfully identified by LC-MS in other tissues (www.uniprot.org; www.proteomicsdb.org) and, except one protein, have also not been detected by another comprehensive MS study of human sperm[51]. Furthermore, we ascertained that protein extraction was complete using harsh extraction conditions followed by protein aggregation capture (PAC)[63], and deep proteomic profiling was ensured by off-line basic reverse-phase chromatography as an additional separation step prior to LC-MS/MS.

## Limit-of-detection (LOD) and copy numbers determined by Parallel Reaction Monitoring (PRM) MS

Considering the depth and reproducibility of our LC-MS/MS analysis, it is worrisome that proteins, previously implied in mammalian sperm signaling by many studies, were not detected in human sperm. Therefore, we applied a targeted MS approach, i.e., PRM-MS, to quantitatively scrutinize whether CFTR, SCNN1A, SLC4A4, and SLC26A6, four signaling proteins that have been invoked in HCO$_3^-$ transport (Fig. 1), are indeed absent from human sperm. To this end, we used dilution series of standard peptides corresponding to these four proteins (Supplementary Table 5) spiked into a mixture of endogenous peptides derived from a defined number of sperm cells. The standard peptides have the same amino-acid sequence and physicochemical properties as their endogenous counterparts, apart from being labelled with stable isotopes to distinguish them from endogenous peptides. Because endogenous and standard peptides elute at the same time, ionize in the same way, and show the same fragmentation in the MS, the difference in their intensities is strictly quantitative. Using this approach, we determined the limit-of-detection (LOD), i.e., the lowest detectable quantity of standard peptides corresponding to CFTR, SCNN1A, SLC4A4, and SLC26A6 in the PRM-MS analysis, in which we targeted both standard peptides and endogenous peptides. We determined the LOD values from four biological replicates with two technical replicates each. Standard peptides were detected in the low attomole range (Fig. 2c, Supplementary Fig. 1a). The LOD values vary between measurements and also between different peptides owing to their different intrinsic properties (Fig. 2d, Supplementary Fig. 1a and Supplementary Data 2). Of note, the minimum number of molecules/cell detected is calculated from the lowest LOD value of a standard peptide obtained from the most sensitive MS measurement. For the different standard peptides, the lowest LOD values ranged from 1 attomole to 50 attomoles, equivalent to 7 and 335 molecules/cell (Fig. 2d). We could not detect the respective endogenous peptides in any PRM measurement, demonstrating that the copy number of CFTR, SCNN1A, SLC4A4, and SLC26A6 must be less than the lowest LOD value.

The absence of these four proteins in our proteome LC-MS/MS analysis (Supplementary Table 2) was not due to technical issues: the respective standard peptides could be detected at low attomole quantities, and PRM analysis of digested proteins from HEK cells using the identical experimental set-up readily identified peptides of SLC4A4 and SLC26A6 (Supplementary Fig. 1b). Thus, intrinsic properties of these proteins and the proteolytically generated peptides do not prevent detection by PRM-MS. In conclusion, the proteins CFTR, SCNN1A, SLC4A4, and SLC26A6 are either absent from human sperm or expressed at low copy numbers.

Subunits of the Ca$^{2+}$ pump PMCA are highly abundant, whereas NCKX or NCX exchangers could not be detected. Thus, PMCA is the primary if not only Ca$^{2+}$ export system. By contrast, in sea urchin

sperm, NCKX density is high, only second to the most abundant GC chemoreceptor, whereas a $Ca^{2+}$ pump is 50-fold less abundant[64]. This comparison testifies to the concept that sperm species employ largely different signaling inventories[6].

Next, we explored the dynamic range between minimum copy numbers defined by LOD values of proteins that were not detected and copy numbers of signaling proteins that were detected. Indeed, the absolute abundance of any protein from the positive list has not yet been reported. To initiate benchmarks, we determined the copy numbers of a few selected proteins by titration with standard peptides including CA2, SLC26A3, CatSper1-4, CatSperB, and Slo3 (Fig. 2e and Supplementary Fig. 2). The number of molecules/cell (mean ± s.d., number of experiments) was $154,100 \pm 67,000$ ($n = 4$) for CA2; $23,100 \pm 15,400$ ($n = 4$) for SLC26A3; $2500 \pm 1800$ ($n = 4$) for CatSper 1; $7100 \pm 2500$ ($n = 4$) for CatSper 2; $5000 \pm 2600$ ($n = 4$) for CatSper 3; $2700 \pm 1300$ ($n = 4$) for CatSper 4; $5400 \pm 1000$ ($n = 4$) for CatSper B; and $5300 \pm 2500$ ($n = 3$) for Slo3.

The precision of PRM analysis using standard peptides primarily depends on the nature of the protein (soluble or membrane-spanning), the choice of suitable standard peptides, the digestion conditions, and the modification of endogenous peptides[65]. Despite using two different standards (QconCAT vs. AQUA peptides) for sperm from two distant species—sea urchin and humans—overall the copy numbers of CatSper subunits are of similar scale (between 2000 and 8000 molecules/cell) (Fig. 2)[64] testifying to the robustness of our quantitative approach. The 3D cryo-electron microscopy structure of murine CatSper shows an equimolar subunit stoichiometry[56], providing a benchmark for the precision of MS to determine subunit stoichiometries in complexes. Finally, a recent EM tomography study of mouse sperm revealed a zig-zag pattern of CatSper dimers along the flagellum axis with a repeat distance of ~5 dimers/100 nm[66] Assuming a quadrilateral arrangement of the CatSper complex[67] and a flagellum length of 50 μm, a copy number of CatSper is calculated to be ~20,000 molecules/cell. This estimated density is about 3–4 times higher than the average number of CatSper subunits determined by MS. Reasons for this difference could be that not all proteins for MS are solubilized from the membrane of sperm. However, we have scrutinized the recovery of proteins/peptides from membranes during our isolation procedure and noted no losses of CatSper units or other proteins (see Methods). Alternatively, the CatSper density probed by tomography may not be uniform along the flagellum. Whatever the reason for this difference might be, both studies report a high density of CatSper channels in the flagellum. Additionally, the copy numbers of two different solute carriers, SLC26A3 in human and SLC9C1 in sea urchin sperm, are also similar (20,000 vs. 40,000 copies/cell, respectively[64]). In conclusion, the copy numbers of three different types of proteins—ion channels, solute carriers, and a CA enzyme—are one to four orders of magnitude higher than the typical range of LOD values (lowest LOD value of 7 molecules/cell vs. 150,000 molecules/cell for CA2).

These results question the presence of several proteins that have been reported to be key players either directly or indirectly in $HCO_3^-$ transport or regulation of the $CO_2/HCO_3^-/H^+$ equilibrium. We recognize the iconoclastic nature of these results and, therefore, discuss here potential pitfalls and the relative weight of each result. Our conclusions rest on several rationales.

First, low copy numbers derived from peptide-specific LOD values are difficult to reconcile with a physiological function. For example, transport capacity should be able to significantly change substrate concentrations within a few seconds. SLC transporters feature transport rates of a few hundred substrate molecules/s; thus, even if present, copy numbers in the range of <10-20 molecules/cell could not substantially change concentrations of substrates such as $HCO_3^-$ or $Cl^-$ that operate in the millimolar range.

Second, the wide dynamic range between such low LOD values vs. the high copy numbers of proteins that serve similar functions

precludes any prominent role of the low-copy-number proteins. Moreover, several known or potential $HCO_3^-$ transporters of the SLC4 and SLC26A families have been proposed to physically interact with CFTR and with each other to form heterodimers, and that $HCO_3^-$ enters sperm via these complexes[18,28,68]. Our data seriously constrain hypotheses about such potential interactions. A case in point is the proposed interaction between SLC4A4 and SLC26A3[18]: the LOD value of SLC4A4 (<10 copies/cell) is ~2,000-times lower than the copy number of SLC26A3 (20,000 copies/cell). Thus, even if SLC4A4 were present, SLC26A3 would be the physiologically dominant $HCO_3^-$ transporter present in human sperm, unless its transport number is several orders of magnitude lower than that of SLC4A4. The large mismatch between LOD values of SLC4A4 and high copy numbers for SLC26A3 also shows that hetero-dimers either do not exist or are too rare to be physiologically relevant in human sperm. The same argument holds for potential interactions between CFTR/ENaC channels and the SLC26A3 transporter. This comparison underscores that the physiological significance of these low-abundance proteins either on their own or in complex with other abundant proteins needs to be revisited.

For additional reasons, CFTR and ENaC channels may be irrelevant for $HCO_3^-$ transport and hyperpolarization of mammalian sperm. It is disputable as to how CFTR or ENaC channels can affect the membrane potential in the presence of the Slo3 channel that features a 10-fold larger single-channel conductance (CFTR vs. Slo3 is 7.9 pS vs. 70 pS)[14] and is up to a 1000-fold more abundant. In sperm of a CatSper[-/-] and Slo3[-/-] double knock-out mouse, no other conductance above the background or leak conductance is detected[69], excluding currents carried by CFTR and ENaC channels also in mouse sperm. In agreement with these physiological results in mouse, we could not detect CFTR and ENaC proteins in human sperm, and, according to our LOD values, maximally between 10 and <40 molecules/cell would be present if any. CFTR has been implied in capacitation of human sperm based on the inhibitor $CFTR_{inh}$-172 (ref. 70) This compound also inhibits the $H^+/Cl^-$ exchange transporter 3 (CLCN3)[71]. Indeed, CLCN3 was identified in our proteome analysis, suggesting that the effect of $CFTR_{inh}$-172 on capacitation might result from its action on CLCN3.

In support of these conclusions, human mutations or targeted disruption in mice of SLC26 members do not cause fertility phenotypes, except for SLC26A3[-/-] and SLC26A8[-/-] mice that suffer from subfertility[22,61,72]. SLC4A4[-/-] mice do not survive breeding age; therefore, the fertility phenotype is not known. These SLC members are expressed elsewhere in the male reproductive tract, where they serve important roles in spermiogenesis[22,61,72]. The SLC26A family is diverse, features various and often unclear substrate specificities, and it serves broad functions[73]. Once the substrate specificity and turnover number of SLC26A3 is known, its contribution to $HCO_3^-$ metabolism relative to $CO_2$ diffusion can be more quantitatively estimated.

Finally, the 'negative' list contains only proteins that abundantly exist in somatic cells, whereas most entries in the 'positive' list are sperm-specific, except for some house-keeping proteins like PKA or $Na^+/K^+$- ATPase. Yet, even those proteins use some subunits that are sperm-specific. This almost perfect all-or-none pattern reflects the historical course of fertilization research. Scientists chose signaling systems from somatic cells and hypothesized that related signaling pathways, furnished with similar molecules, operate in sperm. A case in point are concepts of olfactory and taste signaling in the nose and the tongue, respectively that were adopted as a signpost that guide the study of chemosensation in sperm. Only later was it recognized that sperm employ sperm-specific molecules that are not expressed in somatic cells. We have also failed to detect a plethora of those somatic proteins implied in chemotaxis or thermotaxis of sperm that, however, are not considered here. Our LOD approach holds promise to provide definitive answers as to which molecules are involved in these sensing

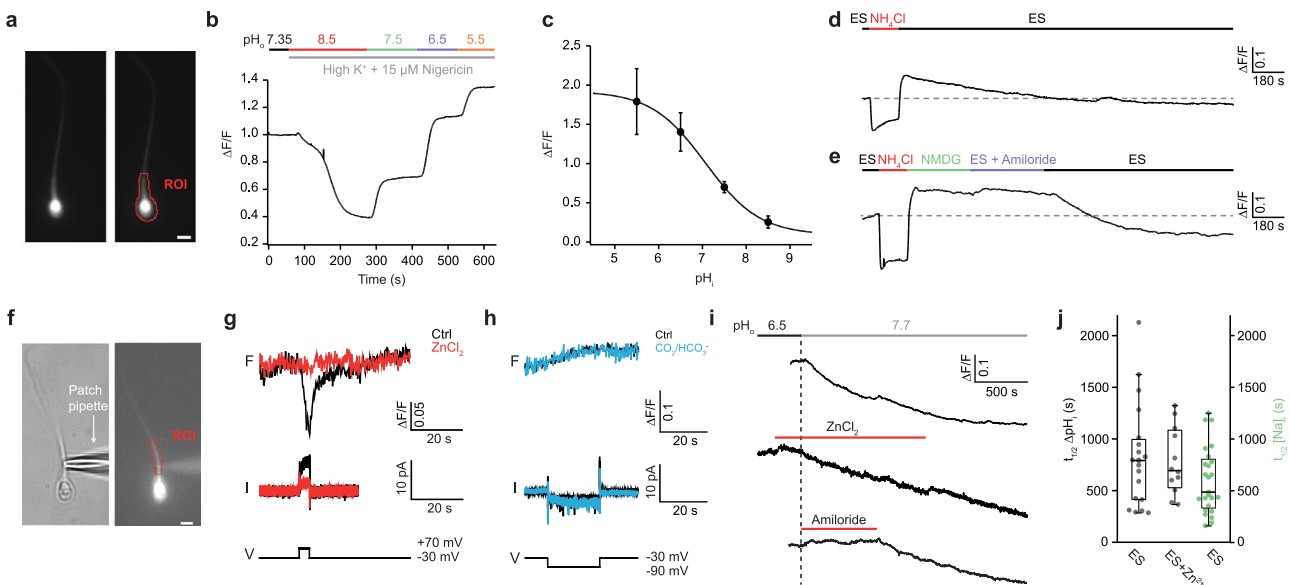

**Fig. 3 | Adjustment of pH$_i$ to changes in extracellular pH (pH$_o$). a** Fluorescence microscopy of a human sperm cell loaded with the pH dye pHrodo Red; ROI, region of interest; scale bar 3.5 μm. **b** Calibration of pHrodo Red with different K$^+$-based pH solutions in the presence of the K$^+$/H$^+$ carrier nigericin (15 μM) and high extracellular K$^+$ (135 mM). **c** Changes in ΔF/F in dependence of pH$_i$ ($n_{exp}$ = 6; $n_{cells}$ = 16; data points and error bars show mean ± SD). **d** Changes in ΔF/F after superfusion of a sperm cell with ES containing NH$_4$Cl (15 mM); the cell first alkalizes, then acidifies ("acid-load") after removal of NH$_4$Cl from the bath, and eventually returns to basal pH$_i$ levels. **e** Sequential perfusion with ES, ES + NH$_4$Cl, ES + NMDG$^+$ instead of Na$^+$, ES +damiloride, and ES. Replacing Na$^+$ with NMDG$^+$ or the presence of amiloride, a Na$^+$/H$^+$ exchange blocker, prevent recovery from the acid load. **f** Patch-clamp fluorometry. Bright-field microscopy (left) and fluorescence image (right) of a human sperm cell loaded with pHrodo Red (12.5 μM); scale bar 3 μm. **g** Changes in pHrodo Red fluorescence (upper), and

current (lower) after stepping voltage from a holding potential $V_m$ = −30 mV to +70 mV to activate H$_v$1 in the absence (black) and presence of Zn$^{2+}$ (100 μM) (red). The pH$_{ES}$ = 7.35; pH$_{pip}$ = 6.5. **h** Changes in fluorescence of pHrodo Red (upper) and current (lower) after stepping voltage from $V_m$ = −30 mV to −90 mV to test the activation of sNHE in the absence (black) and presence of CO$_2$/HCO$_3^-$ (5%/25 mM) (blue). **i** Changes in pH$_i$ upon a switch from pH$_o$ 6.5 to pH$_o$ 7.7 (upper). Zn$^{2+}$ (100 μM) had no effect (middle), whereas amiloride (500 μM) completely prevented alkalization (lower). **j** Half-time t$_{1/2}$ of ΔpH$_i$ (left ordinate) and Δ[Na$^+$]$_i$ (right ordinate for right bar, green) upon change from pH$_o$ 6.5 to pH$_o$ 7.7; Zn$^{2+}$ (100 μM) ($n_{exp}$ = 3, 4, and 5, in presented order). Mann-Whitney test: ES vs. ES+Zn$^{2+}$, $p$ > 0.9; ES (pH signal) vs. ES (Na$^+$ signal), $p$ = 0.1. Box plots represent 25%, 50% (median), and 75% quartiles, with whiskers displaying minimum and maximum data points. Source data is provided as a source data file.

---

mechanisms and anticipate that other proteins from the negative list (Supplementary Table 4) feature similarly low LOD values.

## Recovery from acidification is controlled by amiloride-sensitive Na$^+$/H$^+$ exchange

The MS analysis revealed that components previously implied in HCO$_3^-$ transport are not detectable (except SLC26A3). By readjustment of the CO$_2$/HCO$_3^-$/H$^+$ equilibrium, influx of HCO$_3^-$ or intracellular synthesis of HCO$_3^-$ from CO$_2$, will either alkalize or acidify the cell, respectively. Therefore, we examined changes in pH$_i$ that might provide cues for mechanisms of H$^+$ and HCO$_3^-$ transport. We recorded pH$_i$ in intact single human sperm cells using the fluorescent pH-indicator pHrodo Red (Fig. 3a). The dye was calibrated by exposing sperm to a series of different pH solutions containing the K$^+$/H$^+$ ionophore nigericin and high [K$^+$]$_o$ (Fig. 3b, c). We exposed sperm to conditions previously established to alkalize or acidify cells (Fig. 3d). The dye reported robust rapid alkalization after superfusion of sperm with ES medium containing NH$_4$Cl but no HCO$_3^-$. Removal of NH$_4$Cl triggered a transient acidifying overshoot ("acid-load experiment")[74], before sperm returned slowly to resting pH$_i$. Restoration of pH$_i$ required extracellular Na$^+$: the acidification persisted when ES/NH$_4$Cl was replaced by ES that contained NMDG$^+$ instead of Na$^+$ (to inactivate Na$^+$/H$^+$ exchange), demonstrating that pH$_i$ is adjusted by Na$^+$/H$^+$ exchange (Fig. 3e). Acidification also persisted in the presence of amiloride, a well-known blocker of some Na$^+$/H$^+$ exchangers (e.g., of the SLC9A1 type; Fig. 3e). The pH recovery within 120 s after changing to ES + amiloride vs. ES was significantly different (ΔF/F$_{120s}$ −0.0001 ± 0.008 vs. −0.014 ± 0.013, respectively; paired $t$-test $p$ = 0.001; number of experiments $n_{exp}$ = 4, number of cells $n_{cells}$ = 23). In conclusion, sperm

recover from acidification by amiloride-sensitive Na$^+$/H$^+$ exchange. Because SLC9A1 is absent from sperm, SLC9B1, SLC9B2, and SLC9C1/2 are the remaining candidates for Na$^+$/H$^+$ exchange. However, human SLC9B2 and sea urchin SLC9C1 are insensitive to amiloride[31,75,76], and amiloride sensitivity of SLC9B1 is not known.

We designed several experiments to reveal the contribution of sNHE/SLC9C1 to Na$^+$/H$^+$ exchange. A unique hallmark of sNHE activity in sea urchin sperm, which sets it apart from all other NHEs, is gating by hyperpolarized $V_m$ and modulation by cAMP[31,77]. Voltage gating and cAMP sensitivity rest on a channel-like voltage-sensor motif (VSD) and cyclic nucleotide-binding domain (CNBD) that are absent from all other NHEs[31]. To test for hyperpolarization-triggered Na$^+$/H$^+$ exchange in human sperm, we employed patch-clamp fluorometry (PCF). Sperm were loaded via a patch pipette with the pH dye pHrodo Red (see Methods; Fig. 3f). As a positive control for the pHrodo Red response, we simultaneously recorded proton currents carried by H$_v$1 with the patch-clamp technique and changes in pH$_i$ by pHrodo Red fluorescence (Fig. 3g). Because the voltage regimes of H$_v$1 and sNHE activation do not overlap[31,78], H$^+$ efflux via H$_v$1 cannot interfere with Na$^+$/H$^+$ exchange via sNHE. The cell was held at $V_m$ = −30 mV, close to the resting potential (V$_{rest}$ −35 to −45 mV) of human sperm[79], where sNHE and H$_v$1 are not active. A step from −30 mV to +70 mV, which activates H$_v$1, evoked a proton outward current and a concomitant pH$_i$ increase (Fig. 3g); both signals were abolished by 100 μM Zn$^{2+}$, a potent blocker of H$_v$1 channels[78] (Fig. 3g). Thus, pHrodo Red reliably reports alkalization caused by voltage-gated H$_v$1 activity. By contrast, a 20-s voltage step to −90 mV, which strongly activates sNHE in sea urchin sperm and in a sea urchin sNHE-CHO cell line[31], did not alkalize human sperm; if anything, hyperpolarization caused a slight acidification (Fig. 3h). The

sNHE carries a cAMP-binding domain, and cAMP modulates sNHE activity in sea urchin sperm by a shift of the voltage of half-maximal activation ($V_{1/2}$)[31]. However, even in the presence of $HCO_3^-$, which elevates cAMP in mammalian sperm[9,80], no hyperpolarization-induced alkalization was detected ($pH_i = 7$, $n = 5$; $pH_i = 6.5$, $n = 3$; Fig. 3h). Although the components that directly or indirectly control $pH_i$ may be located in different regions of the flagellum, it can be considered as a single compartment because $H_v1$-induced alkalization rapidly spreads along the flagellum (Supplementary Fig. 3). In summary, no voltage-gated and cAMP-sensitive $Na^+/H^+$ exchange activity exists in human sperm.

sNHE/SLC9C1 is sperm-specific, belongs to the SLC9 family of $Na^+/H^+$ exchangers, and supports classic, reversible $Na^+/H^+$ exchange in sea urchin sperm and cell lines; therefore, it has been regarded as the dominant $pH_i$ regulator in mammalian sperm[32,33]. However, because functionally important residues in the exchanger-, the voltage-sensor, and the cyclic nucleotide-binding domains are missing in mammalian SLC9C1 (ref. 31), $Na^+/H^+$ exchange via SLC9C1 in mammals may be mechanistically and pharmacologically different. The lack of voltage gating is not entirely surprising given that three of four key Arg residues in the S4 voltage-sensor motif of sea urchin SLC9C1 are absent in the mouse and human SLC9C1orthologues[31]. Considering that SLC9C2 is also present in human sperm, we speculate that SLC9C1 and SLC9C2 forms heterodimers that may adopt new properties and mediate amiloride-sensitive $Na^+/H^+$ exchange via a classical alternating-access mechanism[81] rather than by $V_m$ gating and modulation by cAMP. Alternatively, mammalian SLC9C1 may transport substrates other than $Na^+$ or $H^+$. This conclusion is also plausible because a highly conserved Asn-Asp or Asp-Asp motif, which is key for coordination of $Na^+$ ions in $Na^+/H^+$ exchangers[31,76], is lacking in mammalian SLC9C1 (ref. 31) Moreover, SLC9C1[-/-] sperm do not show any alteration in resting $pH_i$[32], something to be expected when $Na^+/H^+$ exchange is abolished. Finally, $Na^+/H^+$ exchange activity of heterologously expressed SLC9C1 is weak if not absent[32]. In this scenario, it is probably SLC9B1 that controls $Na^+/H^+$ exchange in human sperm. Indeed, SLC9B1 predominantly mediates the zona pellucida-induced $pH_i$ increase that is greatly attenuated in SLC9B1[-/-] mouse sperm[35]. Definitive answers are awaited from SLC9B1[-/-]/SLC9B2[-/-] double KO mouse and functional co-expression of SLC9C1/SLC9C2.

A related intriguing feature that remains to be solved is the functional and even molecular interaction between SLC9C1 and sAC. SLC9C1[-/-] sperm are lacking the full-length form of sAC, and motility and infertility phenotypes can be rescued in vitro by cAMP[10,32,35]. Thus, the primary defect of SLC9C1[-/-] sperm may be in cAMP rather than $pH_i$ regulation. The reciprocal control of $Na^+/H^+$ exchange by SLC9C1 and cAMP synthesis by sAC has recently been shown in sea urchin sperm[82]. Non-mammalian sAC is directly activated at alkaline $pH_i$ and not by $HCO_3^-$: alkalization of sea urchin sperm by SLC9C1 stimulates cAMP synthesis[82], and the rise of cAMP modulates the voltage dependence of SLC9C1 activity[31]. Although such reciprocal control is straightforward for a sAC that is controlled by $pH_i$, it is unclear how sNHE controls mammalian sAC, which is activated by $HCO_3^-$ and not $pH_i$[9,82]. Thus, sAC and sNHE/SLC9C1 in human sperm may have adopted new mechanisms of mutual control that remain to be elucidated.

## $CO_2/HCO_3^-$ acidifies rather than alkalizes sperm

The voyage of sperm starts in low-extracellular pH ($pH_o$)/low-$HCO_3^-$ of the epididymis and ends in high-$pH_o$/high-$HCO_3^-$ environs of the oviduct. An increase of intracellular $HCO_3^-$ and $pH_i$ are considered key events during sperm capacitation[83,84], and both have been attributed to $HCO_3^-$ influx via SLC4 and SLC26 transporters or CFTR/ENaC[2,25,27] (Fig. 1, Supplementary Note 1, and Supplementary Table 6). However, these transport systems are not detected by MS, except for SLC26A3 and SLC26A8 (Supplementary Tables 2 and 4). Because pH and $HCO_3^-$ are inextricably coupled by the $CO_2/HCO_3^-/pH_i$ equilibrium, it is

challenging to dissect each contribution to $pH_i$ control. Therefore, we studied how changes in $pH_o$ alone are relayed to the cytosol. Superfusion of sperm was switched from an epididymis-like acidic medium ES/pH 6.5 to an oviduct-like alkaline medium ES/pH 7.7 while recording $pH_i$ fluorometrically. The cytosolic $pH_i$ slowly increased from $pH_i = 6.8 \pm 0.1$ to $7.1 \pm 0.2$ ($n_{exp} = 7$; $n_{cells} = 16$, and $n_{exp} = 6$; $n_{cells} = 21$, respectively, Fig. 3i, see also Fig. 4c) with a half-time $t_{1/2} = 830 \pm 510$ s ($n_{exp} = 3$, $n_{cells} = 20$; Fig. 3i). The alkalization was not significantly affected by $Zn^{2+}$ (Fig. 3i, j) but reversibly abolished by amiloride (Fig. 3i). The alkalization within 120 s after change to pH 7.7 + amiloride vs. pH 7.7 was significantly different ($\Delta F/F_{120s}$ $0.003 \pm 0.014$ vs. $-0.018 \pm 0.013$, respectively; paired Mann–Whitney $p = 0.0001$; $n_{exp} = 2$; $n_{cells} = 16$). We compared the kinetics of $\Delta pH_i$ and intracellular $\Delta[Na^+]_i$ using the $Na^+$-sensitive dye ANG-2. The $t_{1/2}$ and time course of $\Delta pH_i$ ($H^+$ efflux) and $\Delta[Na^+]_i$ ($Na^+$ influx) were not significantly different (Fig. 3j, Supplementary Fig. 4a). This suggests that $H^+$ efflux and $Na^+$ influx is coupled and that under these conditions, ion channels like $H_v1$ and ENaC do not contribute to $H^+$ and $Na^+$ fluxes, respectively. To conclude, amiloride-sensitive $Na^+/H^+$ exchange rather than $H_v1$ alkalize human sperm during transfer from an acidic to an alkaline milieu.

Next, we examined the role of $HCO_3^-$ in $pH_i$ regulation. Previous studies often used an open system in which the aqueous phase containing $HCO_3^-$ and sperm is in contact with air ($pCO_2 = 0.3$ mmHg), whereas cells in the body reside in a quasi-closed system where $pCO_2$ is more than 100-fold higher ($pCO_2 = 40$ mmHg) (Fig. 4a). We emulated the transition from the epididymis to the oviduct by exposing sperm to low and high $pH_i/HCO_3^-$ solutions. Upon switching from ES/$pH_o$ 6.5/5% $CO_2$/3 mM $HCO_3^-$ to ES/$pH_o$ 7.5/5% $CO_2$/30 mM $HCO_3^-$ (quasi-closed system), the $pH_i$ increased slowly from $6.6 \pm 0.1$ to $7.0 \pm 0.1$ (Fig. 4b, c). In the absence of $CO_2/HCO_3^-$, $pH_i$ was in general significantly higher than in the presence of $CO_2/HCO_3^-$. However, the average magnitude and time course of $\Delta pH_i$ and the final $pH_i$ in the presence or absence of $CO_2/HCO_3^-$ were largely similar ($\Delta pH = 0.3$ and 0.4; $t_{1/2} = 830 \pm 510$ and $680 \pm 642$ s, and final $pH_i = 7.1$ and 7.0, respectively) (Fig. 4b–d). Thus, in the quasi-closed system, the presence of $CO_2/HCO_3^-$ does not significantly alter slow intracellular adjustment to alkaline $pH_o$.

By contrast, switching at constant $pH_o = 7.35$ from a $HCO_3^-$-free solution (open system) to a $CO_2/HCO_3^-$-containing solution (from ES to ES/5%$CO_2$/25 mM $HCO_3^-$) evoked a rapid and robust acidification (Fig. 4e, f) that persisted for a 30-min recording time (Supplementary Fig. 4b); the $pH_i$ rapidly returned to basal values after $HCO_3^-$ was removed (Fig. 4e). $CO_2/HCO_3^-$ caused a mean $\Delta pH_i$ of $-0.11 \pm 0.06$ ($n_{exp} = 10$; $n_{cells} = 49$) (Fig. 4g) with a time constant $\tau_{sperm} = 5.6 \pm 2.2$ s ($n_{exp} = 10$; $n_{cells} = 42$) (Fig. 4h). The solution exchange in the chamber, probed by recording the time course of the wash-in of the fluorescent dye Alexa 488, was only slightly faster ($\tau_{Alexa488} = 4.5 \pm 0.6$ s ($n = 4$)) (Fig. 4e), suggesting that solution exchange is partially rate-limiting, and acidification might be even faster. Addition of freshly prepared bicarbonate solution (25 mM $HCO_3^-$) to an open system ($pCO_2 = 0.3$ mmHg) also caused acidification albeit at a slower rate ($\tau_{sperm}$ $9.8 \pm 3.1$ s; $n_{exp} = 4$; $n_{cells} = 20$) (Supplementary Fig. 4c). Influx of $CO_2$ and uptake of $HCO_3^-$ shift the $CO_2/HCO_3^-/H^+$ equilibrium in opposite directions. Therefore, rapid acidification is consistent with $CO_2$ influx. Accordingly, rapid acidification was not significantly affected by inhibitors of CFTR channels, $HCO_3^-$ transporters, or $Na^+/H^+$ exchangers (Fig. 4g, h). Finally, no outward current reflecting $HCO_3^-$ influx via CFTR was observed in PCF recordings (Supplementary Fig. 4d). Although the $pH_i$ recordings do not exclude $HCO_3^-$ influx via SLC26A3 or SLC26A8, the rapid acidification demonstrates that $CO_2$ diffusion across membranes overwhelms $HCO_3^-$ influx via transporters.

Similar experiments using HEK and CHO cells also resulted in robust acidification by $CO_2/HCO_3^-$ (Fig. 4e, f). However, compared to sperm, the acidification kinetics was significantly slower: 2.2- and 9.4-fold, respectively ($\tau_{HEK} = 12.3 \pm 1.6$ s, $n_{exp} = 4$, $n_{cells} = 38$;

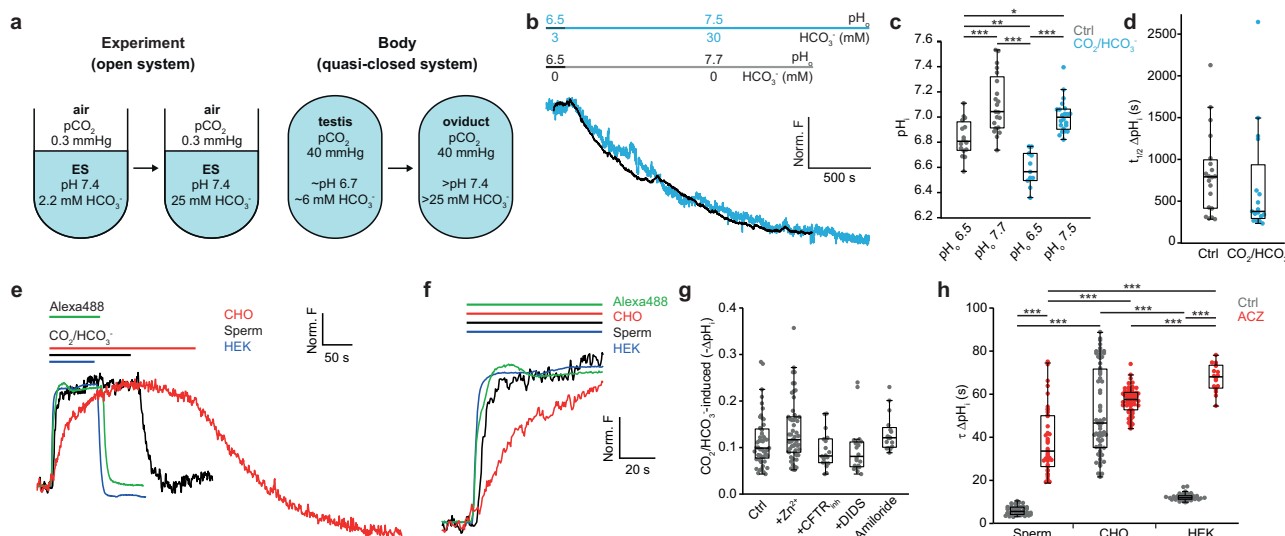

**Fig. 4 | Diffusion of CO₂ across membrane controls pH$_i$ homeostasis. a** Scheme of the difference between an open experimental system in equilibrium with air (pCO₂ = 0.3 mmHg) and a quasi-closed system in equilibrium with body fluid (pCO₂ = 40 mmHg). **b** Time course of alkalization after a switch from pH$_o$ 6.5 to pH$_o$ 7.7 (control, black, same recording as in Fig. 3l) and from pH$_o$ 6.5 to pH$_o$ 7.5 in the presence of CO₂/HCO₃⁻ (5%/3 mM, 5%/30 mM) (blue). **c** Final pH$_i$ after incubating sperm for 30 min at the indicated pH$_o$ in the absence (grey) or presence of CO₂/HCO₃⁻ (pH$_o$ 6.5 + 5%/3 mM or pH$_o$ 7.5 + 5%/30 mM; blue). Two-factor ANOVA followed by post-hoc Tukey's test for multiple comparison: pH$_o$ 6.5 vs. pH$_o$ 7.7 $p < 0.0001$; pH$_o$ 6.5 vs. pH$_o$ 6.5 CO₂/HCO₃⁻ $p = 0.003$; pH$_o$ 6.5 vs. pH$_o$ 7.5 CO₂/HCO₃⁻ $p = 0.014$; pH$_o$ 7.7 vs. pH$_o$ 6.5 CO₂/HCO₃⁻ $p < 0.0001$; pH$_o$ 7.7 vs. pH$_o$ 7.5 CO₂/HCO₃⁻ $p = 0.23$; pH$_o$ 6.5 CO₂/HCO₃⁻ vs. pH$_o$ 7.7 CO₂/HCO₃⁻ $p < 0.0001$. No significant interaction between the factors ($p = 0.09$). **d** Half-time t$_{1/2}$ of ΔpH$_i$ upon a switch from pH$_o$ 6.5 to pH$_o$ 7.7 without (black) and from pH$_o$ 6.5 to pH$_o$ 7.5 with CO₂/HCO₃⁻ (5%/3 mM and 5%/30 mM) (blue; $n_{exp}$ = 3 and 5, respectively). Mann-Whitney test: Ctrl vs. CO₂/HCO₃⁻ $p = 0.08$. **e** Normalized changes in pHrodo Red fluorescence upon switching from pH$_o$ 7.35 to pH$_o$ 7.35/CO₂/HCO₃⁻ (5%/25 mM) solution in sperm (black), HEK293 cells (blue), and CHO cells (red). Green trace mixing kinetics

in the recording chamber measured with Alexa488. **f** Extended time scale of panel (e). **g** HCO₃⁻-induced ΔpH$_i$ upon switching from pH$_o$ 7.35 to pH$_o$ 7.35/CO₂/HCO₃⁻ (5%/25 mM) in the presence of inhibitors. H$_v$1, Zn²⁺ (100 μM); CFTR, CFTR$_{inh}$172 (100 μM); SLC4, DIDS (100 μM); and SLC9A, amiloride (500 μM; $n_{exp}$ = 10, 11, 3, 4, and 4, respectively). Kruskal–Wallis test followed by post-hoc Dunn's test for multiple comparison: Ctrl vs. Zn²⁺, $p = 0.7$; Ctrl vs. CFTR$_{inh}$172 $p > 0.9$; Ctrl vs. DIDS $p > 0.9$; Ctrl vs. amiloride $p = 0.6$. **h** Time constants τ of ΔpH$_i$ upon switching from pH$_o$ 7.35 to pH$_o$ 7.35/CO₂/HCO₃⁻ in the absence (grey) or presence of ACZ (100 μM, red) in human sperm, CHO, and HEK293 cells ($n_{exp}$ = 10, 9, 5, 3, 4, and 2, in presented order). Two-factor ANOVA followed by post-hoc Tukey's test for multiple comparison: sperm vs. sperm ACZ $p < 0.0001$; sperm vs. CHO $p < 0.0001$; sperm vs. HEK $p = 0.2$; CHO vs. HEK $p < 0.0001$; CHO vs. CHO ACZ $p = 0.21$; HEK vs. HEK ACZ $p < 0.0001$; sperm ACZ vs. CHO ACZ $p < 0.0001$; sperm ACZ vs. HEK ACZ $p < 0.0001$; CHO ACZ vs. HEK ACZ $p = 0.009$. Significant interaction between the factors ($p < 0.0001$). Box plots represent 25%, 50% (median), and 75% quartiles, with whiskers displaying minimum and maximum data points. Source data is provided as a source data file.

τ$_{CHO}$ = 52 ± 20 s, $n_{exp}$ = 5, $n_{cells}$ = 64) (Fig. 4e, h). The large difference in surface-to-volume ratio between sperm flagella and CHO or HEK cells cannot account for the vastly different kinetics, because CO₂ equilibrates across membranes within milliseconds (Supplementary Note 2). We reasoned that CA activity, which controls the settling time of the CO₂/HCO₃⁻/H⁺ equilibrium, determines the acidification rate. Indeed, the CA inhibitor acetazolamide (ACZ) markedly slowed down acidification kinetics in sperm and HEK293 but not CHO cells (Fig. 4h), suggesting that CA activity is high in HEK293 and sperm cells but low in CHO cells. Indeed, CA2 is one of the most abundant proteins that feeds into this signaling pathway in human sperm (Fig. 2b) and its density of 154,000 copies/cell is exquisitely high (Fig. 2e). The equilibration times in CHO cells and in aqueous solution are largely similar (52 s vs. 30 s; Supplementary Note 2)[85,86], underscoring that the CA abundance in CHO cells is low. In conclusion, at physiological pCO₂, rapid CO₂ diffusion across sperm membranes and high CA activity controls HCO₃⁻ production and thus pH$_i$.

We also examined voltage-gated and CO₂/HCO₃⁻-evoked pH$_i$ changes in mouse sperm, which lack H$_v$1-mediated proton currents[30] but express the sNHE SLC9C1[33]. Both wt and SLC9C1⁻/⁻ mouse sperm displayed similar behavior: Superfusion with CO₂/HCO₃⁻ also evoked rapid acidification, whereas de- or hyperpolarizing V$_m$ steps did not evoke proton currents or pH$_i$ changes, either before or after HCO₃⁻ treatment (Supplementary Fig. 4e, f). These experiments confirm that H$_v$1 currents are absent in mouse sperm[30] and show that sNHE/SLC9C1

does not serve as a voltage-gated Na⁺/H⁺ exchanger in either human or mouse sperm.

Exposure to higher CO₂/HCO₃⁻ concentrations acidifies sperm, consistent with CO₂ diffusion across membranes followed by readjustment of the CO₂/HCO₃⁻/H⁺ equilibrium and production of H⁺ and HCO₃⁻. Thus, HCO₃⁻ accumulates inside sperm due to rapid CO₂ diffusion across membranes rather than direct import by transporters. We speculate that in previous studies, which report a small alkalization in response to exposure to CO₂/HCO₃⁻ (Supplementary Note 1), sources of error were: (*i*) limited time resolution, which missed initial acidification; (*ii*) use of an open CO₂/HCO₃⁻ system in equilibrium with atmospheric rather than tissue pCO₂; during hour-long capacitation times, sperm slowly alkalize due to an increase in pH$_o$ by CO₂ degassing into the air; and (*iii*) difficulties with pH control of HCO₃⁻ solutions[46].

We estimate that acidification as small as ΔpH$_i$ = -0.15 reflects the production of 10-30 mM H⁺ and, thereby, HCO₃⁻ (see Methods). This concentration matches the HCO₃⁻ sensitivity range of sAC (K$_{1/2}$ = 14 mM)[87]. Thus, small changes in pH$_i$ reflect physiologically relevant changes in HCO₃⁻ concentrations. Cytosolic alkalization has been considered a key step for capacitation However, because HCO₃⁻ and pH$_i$ are inextricably linked by the CO₂/HCO₃⁻/H⁺ equilibrium, it is difficult to gauge the physiological significance of pH$_i$ changes on their own for capacitation. As first physico-chemical principles of CO₂ diffusion across membranes are generally valid[88-90], we anticipate that

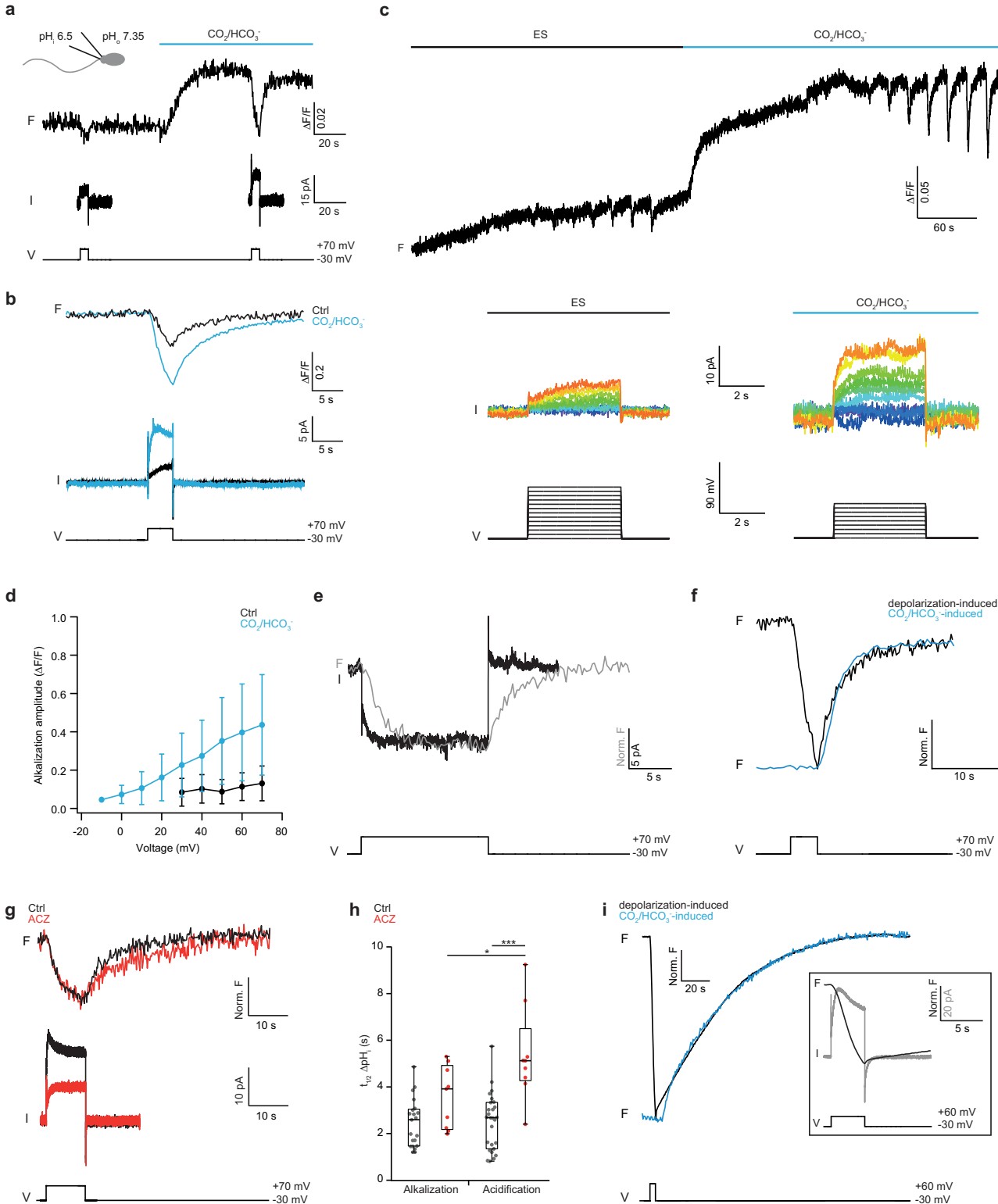

HCO$_3^-$ regulation in sperm of other mammalian species is governed by similar mechanisms.

## Hv1 activation only transiently increases pH$_i$

We next examined how H$_v$1 contributes to pH$_i$ homeostasis by PCF, and how its activity is affected by changes in the CO$_2$/HCO$_3^-$/H$^+$ equilibrium. The mechanism of H$_v$1 activation is unique, as it depends on both V$_m$ and the transmembrane pH difference ($\Delta$pH$_{tm}$ = pH$_o$ − pH$_i$)[78,91]. When $\Delta$pH$_{tm}$ = 0, the threshold voltage V$_{thr}$ at which H$_v$1 activity becomes

noticeable is about +30 mV; when $\Delta$pH$_{tm}$ > 0 or <0, the activation curve is shifted to more negative and positive potentials, respectively[78,91] (Supplementary Fig. 6). Consequently, $\Delta$pH$_{tm}$ sets both the electromotive force $\mathcal{E}$ and the H$_v$1 open probability P$_o$. We reasoned that a positive $\Delta$pH$_{tm}$, i.e., intracellular acidification, caused by CO$_2$/HCO$_3^-$ exposure should markedly enhance H$_v$1 activity. This prediction is borne out by experiment: H$_v$1 currents and associated pH$_i$ signals were greatly increased by CO$_2$/HCO$_3^-$ (Fig. 5a, b). We determined V$_{thr}$ and the voltage dependence of H$_v$1-mediated changes in pH$_i$ with and without

**Fig. 5 | H$_v$1 activity is transiently enhanced by CO$_2$/HCO$_3^-$. a** Patch-clamp fluorometry on pHrodo Red-loaded human sperm. H$_v$1 was activated by a V$_m$ step from −30 mV to +70 mV before and after superfusion of sperm with CO$_2$/HCO$_3^-$ (5%/25 mM) (blue bar; pH$_i$ = 6.5, pH$_o$ = 7.35) while changes in fluorescence (F) and H$_v$1 currents (I) were recorded. **b** Superposition of pHrodo Red fluorescence (upper) and current recordings (lower) under control (black) and CO$_2$/HCO$_3^-$ (blue) conditions. **c** Voltage dependence of pH$_i$ responses (upper) and H$_v$1 currents (middle) before (ES) and after superfusion with CO$_2$/HCO$_3^-$ solution. V$_m$ was stepped for 4 s from −30 mV to +70 mV in increments of 10 mV (lower). **d** Normalized amplitude ΔF/F of voltage-evoked pH$_i$ changes with ES (control, black) and CO$_2$/HCO$_3^-$ (blue). Data points and error bars show mean ± SD ($n_{cells}$ = 7). **e** Superposition of changes in pH$_i$ (grey) and H$_v$1 currents (black) after stepping V$_m$ from −30 mV to +70 mV for 15 s. **f** Superposition of normalized acidification time course after superfusion of sperm with CO$_2$/HCO$_3^-$ (blue) or termination of the V$_m$ pulse (black). **g** Time course of alkalization and acidification (upper) and H$_v$1 current (lower) during a 10 s V$_m$ pulse without (black) and with ACZ (100 μM, red) in the presence of CO$_2$/HCO$_3^-$. **h** Half-times t$_{1/2}$ of H$_v$1-mediated alkalization and acidification for controls (grey) and ACZ (100 μM, red). Data points represent single +70 mV voltage steps of several cells; $n_{cells}$ = 7, 4, 13, and 4 for Ctrl$_{Alk}$, ACZ$_{Alk}$, Ctrl$_{Acid}$, and ACZ$_{Acid}$, respectively. Two-factor ANOVA followed by post-hoc Tukey's test for multiple comparison: Ctrl$_{Alk}$ vs. ACZ$_{Alk}$ $p$ = 0.9; Ctrl$_{Acid}$ vs. ACZ$_{Acid}$ $p$ < 0.0001; Ctrl$_{Alk}$ vs. Ctrl$_{Acid}$ $p$ > 0.9; ACZ$_{Alk}$ vs. ACZ$_{Acid}$ $p$ = 0.03. Significant interaction between the factors ($p$ < 0.0187). **i** Patch-clamp fluorometry in CHO cells expressing the human H$_v$1 channel (CHO-hH$_v$1). Superposition of normalized acidification after superfusion with CO$_2$/HCO$_3^-$ (blue) or termination of the V$_m$ pulse (black). *Inset*: Superposition of changes in normalized fluorescence (black) and current (grey) of a CHO-hH$_v$1 cell during a voltage step from −30 mV to +60 mV. Box plots represent 25%, 50% (median), and 75% quartiles, with whiskers displaying minimum and maximum data points. Source data is provided as a source data file.

CO$_2$/HCO$_3^-$, by stepping V$_m$ from −30 mV to +70 mV in increments of 10 mV (Fig. 5c). Both H$_v$1 currents and pH$_i$ signals increased up to tenfold in the presence of CO$_2$/HCO$_3^-$, and V$_{thr}$ shifted to less positive potentials (Fig. 5c, d). The H$_v$1 enhancement by CO$_2$/HCO$_3^-$ was approximately one order of magnitude larger than previously reported[30]. Of note, the H$_v$1 enhancement can be observed shortly after CO$_2$/HCO$_3^-$ treatment and does not require hour-long capacitating conditions. In conclusion, the CO$_2$/HCO$_3^-$-induced acidification provides a simple, straightforward explanation for the enhancement of proton currents under capacitating conditions by increasing $\mathscr{E}$ and the P$_o$ of H$_v$1, and no other ill-defined capacitation mechanism needs to be invoked.

The rapid acidification after superfusion with CO$_2$/HCO$_3^-$ might be rate-limited by mixing solutions (Fig. 4e, f). To gauge the true response time of the CO$_2$/HCO$_3^-$ buffer system, we perturbed the equilibrium by 15-s voltage pulses. H$_v$1-mediated proton currents activated and deactivated rapidly, whereas the ensuing alkalization and subsequent acidification developed more slowly (t$_{1/2}^{al}$ = 2.5 ± 1.0 s ($n_{cells}$ = 13) and t$_{1/2}^{ac}$ = 2.5 ± 1.2 s ($n_{cells}$ = 7)) (Fig. 5e, h). Of note, pH$_i$ recovered to initial baseline levels without any detectable ionic influx (Fig. 5b), suggesting that re-acidification occurs via electroneutral CO$_2$ diffusion into sperm. By contrast, diffusional exchange of pH buffers and dye molecules between cytosol and pipette proceeds on a minute time scale[92] (Supplementary Fig. 5a; see Methods) and, therefore, cannot account for the observed kinetics of the changes in fluorescence. Moreover, the time course of recovery from alkalization after closure of H$_v$1 channels perfectly matches the time course of acidification by CO$_2$/HCO$_3^-$ superfusion (Fig. 5f). In addition, the CA inhibitor ACZ significantly slowed down recovery from Hv1-induced alkalization (t$_{1/2}^{al}$ = 3.6 ± 1.3 s and t$_{1/2}^{ac}$ = 5.4 ± 2 s) ($n_{cells}$ = 4) (Fig. 5g, h), suggesting that re-acidification after Hv1 closure indeed occurs via CO$_2$ diffusion into sperm and subsequent HCO$_3^-$ synthesis and H$^+$ production catalyzed by CA. Of note, the variance of t$_{1/2}^{ac}$ and t$_{1/2}^{al}$ between sperm cells was larger than that between multiple V$_m$ pulses in the same cell, suggesting that sperm quantitatively differ in their response to a ΔpH$_i$ challenge (Supplementary Fig. 5b).

Similar experiments in CHO cells expressing H$_v$1 (CHO-hH$_v$1) confirmed these results obtained in sperm. H$_v$1 currents rapidly activated, whereas the ensuing alkalization was delayed, slower, and never saturated during a 3-s V$_m$ pulse (Fig. 5i, inset). The recovery, i.e., re-acidification after the voltage pulse, was approximately 10-fold slower compared to sperm (t$_{1/2}^{ac}$ = 32.2 ± 7 s; $n_{cells}$ = 3, Fig. 5i) (Supplementary Fig. 5c, d). Finally, the time courses of acidification when challenged either with CO$_2$/HCO$_3^-$ or after a V$_m$ pulse were similar (Fig. 5i). The large difference of pH$_i$ kinetics in sperm vs. CHO cells after V$_m$-induced perturbations reiterates the notion that CA in sperm is highly abundant compared to CHO cells. Collectively, these experiments show that a rapid pH$_i$ buffer system is achieved by CO$_2$ diffusion in and out of sperm followed by readjustment of the CO$_2$/HCO$_3^-$/H$^+$ equilibrium.

Depolarization-induced activation of H$_v$1 transiently alkalizes the cytosol; however, upon repolarization, Hv1 closes, and the CO$_2$/HCO$_3^-$ buffer system rapidly restores the initial pH$_i$. The high turnover number of CA ($0.5$-$1.0 \cdot 10^6 \, \text{s}^{-1}$)[93–95] together with the high CA2 copy number of 154,000 molecules/cell renders the CO$_2$/HCO$_3^-$/H$^+$ equilibrium a powerful buffering system that rapidly responds to changes in pH$_i$ and HCO$_3^-$ concentration (Fig. 5). The high copy number of CA may be beneficial for another reason: it greatly enhances CO$_2$ flux across membranes by removal of CO$_2$ from unstirred layers[96]. Both CA4, attached to the extracellular site of the membrane, and intracellular CA2 may facilitate CO$_2$ diffusion. Indeed, CO$_2$ perfusion accelerates the flagellar beat of sperm from mouse and humans, a clear sign of cAMP increase, and in CA4$^{-/-}$ mice the CO$_2$-induced response is blunted[36]. Thus, the rapid establishment of the CO$_2$/HCO$_3^-$/H$^+$ equilibrium by CA is centre stage of sperm metabolism.

Ever since its discovery, the function of H$_v$1 in human sperm has remained enigmatic. H$_v$1 has been proposed to represent the long-sought molecule that can cause alkalization under capacitating conditions[30] and, at the same time, hyperpolarization due to H$^+$ efflux; both processes are considered important steps during capacitation (for a comprehensive review, see[2]). Although the positive V$_m$ values required for H$_v$1 activation in vitro are unlikely to prevail in sperm[69,97], the presence of proteolytically cleaved H$_v$1 isoforms in human sperm[39] and albumin that binds to H$_v$1 (ref. 98) shift the V$_{1/2}$ to more favorable potentials. In light of the substantial H$^+$ production by HCO$_3^-$ synthesis and H$_v$1's unique regulation by both V$_m$ and ΔpH, we suggest that H$_v$1 curtails strong acidification during HCO$_3^-$ synthesis from CO$_2$ and, thus, sustains HCO$_3^-$ synthesis. It is in this sense that H$_v$1 is required for capacitation rather than promoting alkalization per se. The dependence of H$_v$1 on both V$_m$ and ΔpH$_{tm}$ (ref. 39,91) implies that H$_v$1 opens only at V$_m$ values positive to the Nernst potential for protons, i.e., H$_v$1 can carry only proton outward currents (Supplementary Fig. 6). This mechanism is self-limiting: when H$_v$1 is open, the resulting hyperpolarization and the decrease of ΔpH$_{tm}$ due to H$^+$ efflux pushes V$_{thr}$ to more positive values and H$_v$1 closes again. Indeed, on repolarization and subsequent H$_v$1 closure, the original pH$_i$ was rapidly restored by the CO$_2$/HCO$_3^-$ buffer system (Figs. 3g and 5). Therefore, H$_v$1 is ill-suited to permanently alkalize sperm. Instead, H$_v$1 may serve as an auto-regulatory, outwardly-rectifying H$^+$ valve that, by design, prevents intracellular acidification. H$_v$1 serves a similar function in macrophages during production of reactive oxygen species[99,100]. A corollary of this mechanism is that, depending on ΔpH$_{tm}$, a change in V$_m$ is not strictly required to activate H$_v$1 (Supplementary Fig. 6). In support of this new function, H$_v$1 rapidly blunts pH$_i$ bursts caused by HCO$_3^-$ production, and the H$_v$1-mediated pH$_i$ change is rapidly reversed via the CO$_2$/HCO$_3^-$/H$^+$ buffer system after H$_v$1 closure (Fig. 5). Future work needs to address why human sperm require H$_v$1 and two different Na$^+$/H$^+$ exchangers to expel H$^+$ from the flagellum, whereas in mice sperm, which lack H$_v$1, Na$^+$/H$^+$ exchange seems to be sufficient.

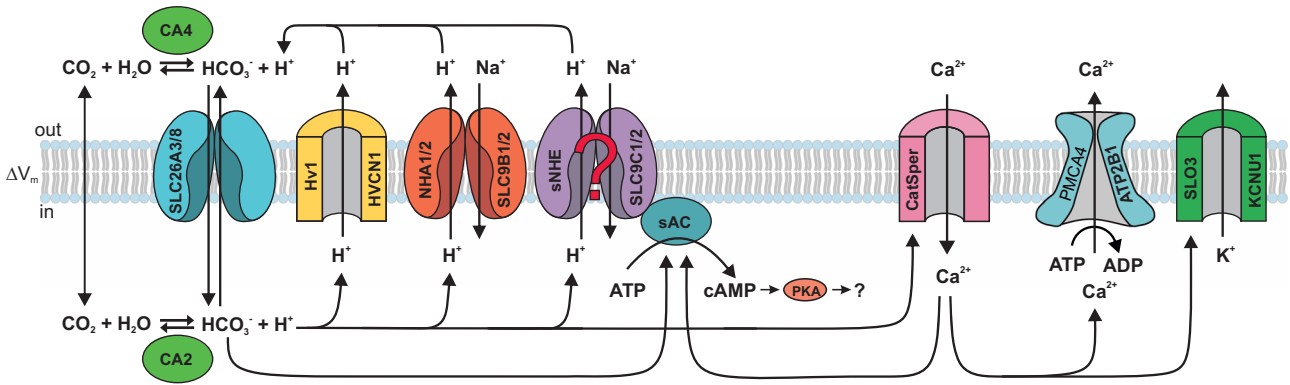

**Fig. 6 | Scheme of pHᵢ, HCO₃⁻, and Ca²⁺ control in human sperm.** Signaling components that have been established by MS or functional experiments. Question mark indicates that the substrate transport of SLC9C1/2 has not yet been established. The figure was designed by Dr. René Pascal.

In summary, the $CO_2/HCO_3^-/H^+$ system represents a central hub of cellular signaling in human sperm and we identify key components that control this chemical equilibrium. $CO_2$ enters sperm by diffusion across the cell membrane and reacts with $H_2O$ to form $HCO_3^-$ and $H^+$. To sustain synthesis of $HCO_3^-$ from $CO_2$ requires that the ensuing acidification is continuously blunted either by $Na^+/H^+$ exchange via SLC9B or SLC9C, or $H^+$ extrusion via $H_v1$, or both. Although our findings contest decades-old concepts of pHᵢ and $HCO_3^-$ regulation in human sperm, they simplify the network of cellular reactions or components (Fig. 6), and may advance the study of mechanisms underlying male infertility and the development of new contraceptives.

Mass spectrometry is commonly employed to identify proteins unequivocally; here, we use MS to question several proteins from playing a role in human sperm physiology. Because we define in rigorous quantitative terms the sensitivity limit of peptide detection, this approach narrowly restricts alternative interpretations and holds promise to unequivocally answer the most basic question: is a protein there, or not? Thereby, it may help to clarify decade-old issues and conundrums.

## Methods
Our research complies with all relevant ethical regulations. The committees and institutions involved are named at the respective places below.

### Sperm preparation
Human semen samples were donated by healthy adult caucasian males (age: 21–56 years) with their prior written consent and the approval of the ethic committee of the University of Bonn (042/17). Handling and experiments with human semen samples were performed in agreement with the standards set by the Declaration of Helsinki. Ejaculates were liquefied at room temperature (RT, 20-23 °C) for 30–60 min and purified by the "swim-up" procedure[43] in human tubular fluid (HTF) medium containing (in mM): NaCl 93.8, sodium-lactate 21.4, HEPES 21, KCl 4.69, sodium bicarbonate (NaHCO₃) 4, D-glucose 2.78, CaCl₂ 2.04, sodium pyruvate 0.33, MgSO₄ 0.2, and potassium dihydrogen phosphate (KH₂PO₄) 0.37 (adjusted to pH 7.35 with NaOH) and 5% $CO_2$. After 30-45 min, motile sperm were collected from the supernatant, centrifuged at 700 xg for 10 min and resuspended in fresh HTF. Samples were stored in HTF at 37 °C diluted to 1*10⁷ cells/ml until they were further prepared for recordings.

Mouse epididymis were obtained from C57Bl/6 N WT or sNHE-KO mice that were anaesthetized with isoflurane (Abbvie, Ludwigshafen, Germany) and killed by cervical dislocation according to the German law of animal protection and the district veterinary office (Landesamt für Natur, Umwelt und Verbraucherschutz Nordrhein-Westfalen, LANUV). sNHE knockout mice were purchased from the Jackson Laboratory (B6; 129S6-Slc9a10tm1Gar/J, stock number: 007661). Mice were housed at 21 °C with 55% relative humidity at a 12 h light/dark cycle.

For mouse sperm preparation, cauda and corpus epididymides were excised and separated from fat. The tissue was incised several times and transferred to modified TYH medium containing (in mM): NaCl 138, sodium-lactate 10, HEPES 10, glucose 5.6, KCl 4.8, CaCl₂ 2.0, KH₂PO₄ 1.2, MgSO₄ 1.0, sodium-pyruvate 0.5 (adjusted to pH 7.4 with NaOH) for the "swim-up" procedure. After 15 min, sperm were collected from the supernatant, centrifuged at 700 xg for 10 min and resuspended in fresh TYH medium. Sperm were diluted to 1*10⁷ cells/ml in TYH and stored at 37 °C until further processed.

For single-cell fluorometry, cells were loaded with 3.3 μM pHrodo-Red-acetoxymethyl ester (pHrodo-Red-AM, Thermo Fisher Scientific, Waltham, MA) for 12 min, 2 μM Calbryte520-AM (AAT Bioquest, Sunnyvale, CA) and 0.05% Pluronic F-127 for 60 min, or 10 μM Asante Natrium Green (ANG2, TEFLabs, Austin, TX) and 0.05% Pluronic F-127 for 60 min, at 37 °C for pH, calcium, and sodium measurements, respectively, followed by centrifugation (700 x g for 10 min) and washing.

### Digestion of proteins from human spermatozoa cells for MS analysis
Proteins from $1–5 \times 10^6$ sperm were extracted and subjected to in-solution digestion as follows. The sample was diluted two-fold by addition of an equal volume of human sperm lysis buffer (8% (w/v) sodium dodecyl sulfate (SDS), 10 mM ethylenediaminetetraacetic acid (EDTA), 10 mM ethylene glycol-bis(2-aminoethylether)-N, N, N', N'-tetraacetic acid (EGTA), 20 mM dithiothreitol (DTT), and 100 mM NH₄HCO₃), and boiled for 10 min at 99 °C. The sample was sonicated in a pre-warmed (15 °C) sonication device (Bioruptor® Plus, Diagenode, Seraing, Belgium) for 10 min with alternating cycles of 30 s ON, followed by 30 s OFF. After sonication, 50 mM NH₄HCO₃ buffer (pH 8) was added resulting in a final SDS concentration of 1%, and the sample was incubated with 500 units of universal nuclease (Pierce, 250 U/μl) in the presence of 1 mM MgCl₂ for 30 min at 37 °C. Proteins were reduced with 10 mM DTT for 30 min at 60 °C and then alkylated with 40 mM iodoacetamide (IAA) for 30 min at 37 °C, followed by another incubation step with 20 mM DTT for 5 min at 37 °C. SP3 beads were prepared as described and 3 mg of SP3 beads were added to the reduced and alkylated proteins, followed by addition of 100% ethanol (LiChrom grade) to a final concentration of 50% (v/v) ethanol and incubation for 5 min at 24 °C with shaking at 1000 rpm in a ThermoMixer® C (Eppendorf, Hamburg, Germany). SP3 beads were washed three times with 80% (v/v) ethanol using an in-house made magnetic rack. After the final washing step, SP3 beads were resuspended in 50 mM NH₄HCO₃ buffer. Rapigest (Waters, Eschborn, Germany) was added to a final concentration of 0.1% (w/v) and trypsin (sequencing grade,

Promega, Walldorf, Germany) was added in 1/20 (w/w) trypsin to protein ratio for protein digestion overnight at 37 °C. Digestion was stopped by acidifying the sample with 100% formic acid (FA) so that the pH of the sample was between 2 and 3 and incubated for 30 min at 37 °C, followed by centrifugation for 10 min at 4 °C and 17,000 xg. Using a magnetic rack, the peptide containing supernatant was transferred to another reaction tube and 80% (v/v) acetonitrile (ACN)/ 0.1% (v/v) formic acid (FA) were added to the supernatant yielding a concentration of 4.5% ACN, pH 2-3. The peptide mixture was cleaned up using a C18 spin column (Harvard Apparatus, Holliston, MA) according to the manufacturer's protocol and dried in a SpeedVac concentrator. For whole-proteome LC-MS/MS analysis, dried peptides were reconstituted in 50 µl (containing peptides derived from $1-5 \times 10^6$ sperm cells) 10 mM $NH_4OH$, pH 10 for further separation by basic reverse-phase chromatography (see below). For PRM-MS analysis (see below), peptide samples derived from sperm were aliquoted in amounts containing peptides derived from 180,000 sperm cells and dried in a SpeedVac concentrator.

To control for complete solubilization and protein recovery, human sperm cells were denatured as described above and then in an additional step centrifuged at 30,000 xg (26,000 rpm, S100-AT4 rotor, UZ MX150, Thermo Sorvall) to separate insoluble material. The residual small pellet was again dissolved in 8% (w/v) SDS, digested, and analyzed by LC-MS. Furthermore, to control for lost peptides during enzymatic digestion, SP3 beads were—after the above-described elution steps of peptides—incubated with 50% (v/v) ACN and 0.1% (v/v) FA for 15 min at 37 °C. The solution containing eventually residual extracted peptides was dried in a SpeedVac concentrator and then analyzed by LC-MS as above.

### Digestion of proteins from HEK293 cells for control PRM-MS
HEK293 cell lysate containing 500 µg of protein was directly subjected to in-solution digestion starting with nucleic acid digestion in the presence of 50 mM $NH_4HCO_3$ buffer (pH 8), 500 units of universal nuclease and 1 mM $MgCl_2$. Further sample processing was performed exactly as described for sperm sample above. The peptide mixture was stored in 10 µg aliquots at −20 °C.

### Off-line basic reverse-phase high-performance liquid chromatography (bRP-HPLC)
Off-line by bRP-HPLC was performed on an Agilent 1100 series HPLC system. Peptides derived from digestion of $1-5 \times 10^6$ sperm cells were dissolved in 50 µl mobile phase A (10 mM $NH_4OH$ in $H_2O$, pH 10) and injected. Elution was performed at a flow rate of 60 µl/min using mobile phase A and B (10 mM $NH_4OH$ in 80% ACN, pH 10) with a Waters XBridge C18 column (3.5-µm particles, 1.0 mm inner diameter, and 150 mm in length). The gradient was 5% B for 5 min, then changed to 10% B in 3 min, to 35% B in 34 min, to 50% B in 8 min, to 90% B in 1 min, to 95% in 5 min, back to 5% B in 2 min, then held at 5% B for 6 min (64 min total runtime). The elution was recorded at 214 nm. 60 µl fractions including the flow-through were collected. Fractions were combined into total 24 fractions, dried in a SpeedVac concentrator, and stored at −20 °C for subsequent LC-MS analysis.

### LC-MS/MS analysis
Dried peptides in each bRP-HPLC fraction were dissolved in 4% (v/v) ACN/0.1% (v/v) TFA in $H_2O$. 24 individual fractions were analyzed on Orbitrap Exploris™480 mass spectrometer (Thermo Fisher Scientific) or on Q Exactive HF-X Hybrid Quadrupole-Orbitrap Mass Spectrometer (Thermo Fisher Scientific), coupled to a Dionex UltiMate 3000 UHPLC system (Thermo Fisher Scientific) equipped with an in-house-packed C18 column (ReproSil-Pur 120 C18-AQ, 1.9 µm pore size, 75 µm inner diameter, 30 cm length, Dr. Maisch HPLC, Ammerbuch, Germany). Mass spectrometers were operated using the software Thermo XCalibur 4.4.16.14 for the Orbitrap Exploris™480 mass spectrometer

and Thermo XCalibur 4.0.27.19 for the Q Exactive HF-X Hybrid Quadrupole-Orbitrap Mass Spectrometer. Samples were separated applying the following 118 min gradient: mobile phase A consisted of 0.1% (v/v) FA in $H_2O$, mobile phase B of 80% (v/v) ACN/0.08% FA (v/v) in $H_2O$. The gradient started at 5% B, increasing to 8% B within 10 min, followed by 8–35% of B within 90 min and by 35% to 55% within 7 min. Finally, the mobile phase was set to 90% of B for 4 min. After each gradient, the column was again equilibrated with 5% B for 6 min. The flow rate was 300 nl/min. Eluting peptides were analyzed in positive mode using a data-dependent acquisition method with a time between master scans of 3 s. MS1 spectra were acquired with a resolution of 120,000 full width at half maximum (FWHM) in the Orbitrap covering a mass range of 350–1600 $m/z$ (data type−profile). Injection time was set to 25 ms and automatic gain control (AGC) target to 300% ($100\% = 1 \times 10^6$). Dynamic exclusion covered 25 s and intensity threshold was set to $5 \times 10^3$. Precursor ion charge state screening was enabled, all unassigned charge states were rejected and only precursors with a charge state of 2-6 were included. MS2 spectra were recorded with a resolution of 15,000 FWHM in the Orbitrap (data type − centroid), injection time was set to 32 ms, AGC target to 75% ($100\% = 1 \times 10^5$) and the isolation window to 1.4 $m/z$. Fragmentation was enforced by higher-energy collisional dissociation (HCD) at 28%.

### Sample preparation for absolute quantification
For the absolute quantification of CA2, SLC26A3, CatSper1-4, Cat-SperB, and Slo3, and the determination of LOD values for SLC4A4, SLC26A6, SCNN1A, and CFTR, isotope-labelled standard peptides from Thermo Fisher Scientific and for analysis of CatSperB, isotope-labelled standard peptides from JPT (Berlin, Germany) were used (Supplementary Table 5). Peptides from both providers were received in lyophilized form. Standard peptides from Thermo Fisher Scientific were dissolved in 40% (v/v) ACN/0.1% (v/v) trifluoroacetic acid (TFA) in $H_2O$, the different peptides were mixed at equimolar concentrations of 1 nanomol, split into aliquots of 10 pmol, dried in a SpeedVac concentrator and stored at −20 °C. Standard peptides from JPT were solubilized in 80% (v/v) 100 mM $NH_4HCO_3$ buffer and 20% (v/v) ACN in $H_2O$. The JPT peptides required an additional endoproteinase digest to remove the Q-tag. For this, 500 pmol of each peptide were mixed, and 0.5 µg trypsin was added to the peptide mixture. Digestion was performed at 37 °C for 5 h, as instructed by JPT. The reaction was quenched with 5 µl of 100% (v/v) FA. The peptides were divided into 10 pmol aliquots, dried in the SpeedVac and stored at −20 °C.

For each quantification experiment, one of the 10 pmol aliquots of heavy peptide mixture was used. Mixtures of Thermo or JPT standard peptides were spiked into the samples at different concentrations, so that final concentrations for Thermo standard peptides in 7 µl of injected sample were (in fmol): 0.001, 0.005, 0.01, 0.05, 0.1, 0.5, 1.0, 5.0, and 10.0. For quantitative analysis with the JPT standard peptides, the following final concentrations in 7 µl of injected sample were used (in fmol): 0.1, 0.5, 1.0, 5.0, 10.0, and 50.0. To reach the aforementioned absolute amount of standard peptides in 7 µl sample, the biological sample (peptides from 180,000 sperm cells, see above), was dissolved in 14 µl or 21 µl of the respective standard peptide dilutions. By this, 7 µl that were injected into the mass spectrometer contained endogenous peptides derived from either 90,000 or 60,000 sperm cells and the specified amount of standard peptide.

For the positive control with peptides derived from HEK293 cells, concentrations of standard peptides were prepared as follows (in fmol): 0.5, 1.0, 5.0, 10.0 and 20.0 of peptide in 7 µl of MS loading buffer. To each aliquot of HEK293 peptides, 70 µl of standard peptide dilution was added yielding 1 µg of peptides in 7 µl.

### Absolute quantification with PRM-MS
For absolute quantification, 7 µl of injected samples were measured on an Orbitrap Exploris™480 mass spectrometer (Thermo Fisher

Scientific) equipped as above. Peptides were separated using a 58 min 0–42% ACN (v/v)/0.1% (v/v) in $H_2O$ gradient and analyzed via an acquisition method involving an MS1 survey scan and targeted MS2 scans. For MS1 scans, the Orbitrap resolution was set to 120,000 FWHM, the scan range ($m/z$) was adjusted to the $m/z$ range of the peptides to be detected, a standard AGC target was used and data was acquired in positive mode. For the targeted MS2 scan, the isolation window ($m/z$) was set to 1.4 $m/z$, a normalized collision energy of 30% and a normalized AGC target of 100% was used, the maximum injection time was adjusted to 100 ms, the Orbitrap resolution was set to 60,000 FWHM, and data was acquired in positive mode. For targeting, the mass and charge state of the peptides as well as the retention time windows were defined in the PRM method settings in the mass spectrometer. The retention time windows were newly determined for each set of measurements.

## Analysis of MS data
Raw files were processed by MaxQuant (MQ) software (version 1.6.5.0)[101,102] with its built-in Andromeda peptide search engine[103]; the following settings were used to identify proteins: trypsin/P was used as digestion enzyme with maximal two missed cleavage sites; MS2 spectra were searched against the UniProt complete *Homo sapiens* proteome sequence database (downloaded in June 2020); carbamidomethylated cysteines were set as fixed, oxidation of methionine and N-terminal acetylation as variable modifications (maximal allowed number of modifications per peptide was set to five); maximum false discovery rate was set to 0.01 both on peptide and protein levels. One peptide with a minimum length of seven amino acids was required for protein identification.

To generate a label-free absolute quantification intensity matrix, the "iBAQ" built-in option in the MaxQuant software was selected without further changes of the default settings. iBAQ intensities calculations use the approach taken in the iBAQ method[60].

The PRM data was analyzed with Skyline (MacLean et al., 2010, version 21.1.0.278, MacCoss Lab Software, University of Washington, Seattle, WA). The mass accuracy (delta mass ≤ 10 ppm) of the transitions was used as cut-off filter to reduce the number of false positives. The Skyline program was provided with the peptide sequences; in the peptide settings section, the isotope modifications for $^{13}$C, $^{15}$N-labelled lysine and arginine were added; in the transition settings section, the instruments and measurement details were provided to the Skyline software. The MS raw files were imported into Skyline, and retention times of the different LC-MS runs were aligned. Transitions of endogenous peptides were compared to signals of labelled peptides; transitions from endogenous peptides were only considered when they displayed the same retention time and fragmentation pattern as the labelled peptides. Only peaks/signals were included in quantification calculations that fulfilled the following criteria: a minimum of three transition ions, minimum of three to five acquired data points. The total integrated peak area was exported from Skyline for both endogenous and standard peptides. The signals from the standard peptides were used to generate calibration curves, which were used to calculate the molar amounts of endogenous peptides. From that, the molecules/cell was calculated. For plotting, data from all peptides corresponding to the same protein were merged (Microsoft Excel 2019, R programming language (version 4.2.2), R Studio, version 2022.07.2). Data displayed as box plots represent 25%, 50% (median), and 75% quartiles, with whiskers displaying minimum and maximum data points, except for outliers outside of 1.5-times interquartile range (IQR).

## Solutions for electrophysiological and fluorometric recordings
For fluorometric and electrophysiological recordings, human and mouse sperm were mixed with 3 mg/ml human serum albumin (HSA) or bovine serum albumin (BSA), respectively, and pipetted on poly-L-lysine coated coverslips submerged in extracellular recording solution

(ES), containing (in mM): NaCl 140, mannitol 10, KCl 5.4, HEPES 5, $CaCl_2$ 1.8, $MgCl_2$ 1.0 (adjusted to pH 7.35 with NaOH). For "acid-load" experiments, ES with $NH_4Cl$ contained (in mM): NaCl 125, $NH_4Cl$ 15, mannitol 10, KCl 5.4, HEPES 5, $CaCl_2$ 1.8, $MgCl_2$ 1.0 (adjusted to pH 7.35 with NaOH); ES with N-methyl-D-glucamine ($NMDG^+$) contained (in mM): $NMDG^+$ 140, mannitol 10, KCl 5.4, HEPES 5, $CaCl_2$ 1.8, $MgCl_2$ 1.0 (adjusted to pH 7.35 with HCl). For pH calibration high-$K^+$ calibration solutions contained (in mM): KCl 135, HEPES/MES/Tris 20, $CaCl_2$ 1.8, $MgCl_2$ 1.0, Nigericin 0.015 (adjusted to pH 5.5, 6.5, 7.5, or 8.5 with HCl or KOH). The ES/pH7.35/$CO_2$/$HCO_3^-$, which was always used when cells were exposed to $CO_2$/$HCO_3^-$ unless otherwise mentioned, contained (in mM): NaCl 115, $NaHCO_3$ 25, mannitol 10, KCl 5.4, HEPES 5, $CaCl_2$ 1.8, $MgCl_2$ 1.0 (adjusted to pH 7.35 with NaOH). The pH conditions of the epididymis (pH 6.5) and oviduct (pH 7.5) were mimicked with the following solutions. ES/pH6.5/ $CO_2$/$HCO_3^-$ contained (in mM): NaCl 137, $NaHCO_3$ 3, mannitol 10, KCl 5.4, MES 5, $CaCl_2$ 1.8, $MgCl_2$ 1.0 (adjusted to pH 6.5 with NaOH); ES/pH 7.5/$CO_2$/$HCO_3^-$ contained (in mM): NaCl 110, $NaHCO_3$ 30, mannitol 10, KCl 5.4, HEPES 5, $CaCl_2$ 1.8, $MgCl_2$ 1.0 (adjusted to pH 7.5 with NaOH). $HCO_3^-$-containing solutions were bubbled with Carbogen (5% $CO_2$, 95% $O_2$) for at least 30 min before the pH was adjusted. $HCO_3^-$-containing solutions were continuously bubbled with Carbogen during experiments to maintain constant pCO$_2$ and pH. Prior to recordings, the pH of all solutions was checked with a calibrated pH-meter (SevenEasy, Mettler Toledo, Columbus, OH) and adjusted, if necessary. All extracellular solutions had osmolarities of 290-300 mOsm/l.

For patch-clamp recordings of mouse sperm, the pipette solution contained (in mM): aspartic acid 130, Mg-ATP 4, NaCl 1.4, MES 1.0, EGTA 1.0 (adjusted to pH 6.5 with CsOH; osmolarity adjusted to 300 mOsm/l with mannitol). For all other cell types, the intracellular solution contained (in mM): $NMDG^+$ 140, Mg-ATP 4.0, NaCl 1.4, EGTA 1.0, MES 1.0 (adjusted to pH 6.5 with methane sulfonate; osmolarity was adjusted to 300 mOsm/l with mannitol). Prior to recording, 12.5 μM pHrodo-Red-maleimide was added.

All chemicals and inhibitors were purchased from Carl Roth (Karlsruhe, Germany), Sigma Aldrich (St. Louis, MO), or Thermo Fisher Scientific (Waltham, MA) if not stated otherwise.

## Experimental setup
Fluorometry and patch-clamp fluorometry (PCF) experiments were performed on an inverted IX71 microscope (Olympus, Tokyo, Japan) equipped with a 60× water immersion objective and an additional 1.6x magnification lens for PCF. Fluorescent indicators were excited with LED light (Spectra X light engine, Lumencor, Beaverton, OR). Excitation and emission filters used were 543/22 nm and 568LP nm for pHrodo-Red and 472/30 nm and 525/35 nm for CalBryte-520 and ANG-2. An EMCCD camera (iXon Ultra DU-897U, Andor Technology, Belfast, Ireland) controlled by the Andor Solis software was used for imaging.

An Axopatch 200B amplifier connected to Digidata 1440 A acquisition board controlled by the software ClampEx (Molecular Devices, Union City, CA) was used for electrophysiological recordings. Patch pipettes were pulled from borosilicate capillaries (Hilgenberg, Malsfeld, Germany) using a DMZ puller (Zeitz Instruments GmbH, Martinsried, Germany). All experiments were conducted at RT.

## Fluorometry
For fluorometric imaging, coverslip areas with non-overlapping cells were chosen. Motile sperm with only their head attached to the coverslip were chosen for analysis. ROIs were drawn by hand and restricted to the sperm head, the midpiece, and up to the first quarter of the principal piece (Fig. 3a). These parts were little perturbed by flagellar beating. A sampling rate of 1 or 2 Hz (0.5 Hz for Supplementary Fig. 4b) was used for imaging. The rest of the sperm flagellum, beating at frequencies around 5 Hz[104], was excluded from analysis. During the whole experiment, a continuous flow of perfusion was

applied and minimized motion of sperm head and midpiece. Depending on the density of the sperm cells on the cover slip, one to several sperm cells could be imaged simultaneously. Number of experiments ($n_{exp}$) denotes the number of cover slips recorded in imaging experiments. Number of cells ($n_{cells}$) denotes the total number of cells imaged.

### Patch-clamp fluorometry (PCF)

For patch-clamp experiments, cells sufficiently distant from other cells were chosen. The initial resistance of the pipettes was 11–16 MΩ for sperm and 3–8 MΩ for HEK293 or CHO cells. Pipette offset and capacitance was compensated before sealing. After Giga-seal formation, the whole-cell configuration was achieved by manually applying short pulses of negative pressure at a holding potential $V_m = -30$ mV. During the break-in process, loading of cells with pHrodo-Red maleimide was observed with the EMCCD camera. The half-time $t_{1/2}$ of filling sperm flagella with dye was about 50 s (Supplementary Fig. 5a) which is about tenfold slower than the reported filling time of mouse sperm with Lucifer Yellow[12]. Whole-cell currents from all cell types were sampled with 2 kHz and images were acquired at 5 Hz. Before and between recordings, cells were held at −30 mV. Cells were continuously perfused with extracellular solution. Motile sperm with only their head attached to the coverslip and a visible cytoplasmic droplet were chosen for PCF experiments. ROIs were drawn by hand and were restricted to around 10-15 μm in length, including the midpiece and the first part of the principal piece (Fig. 3f).

### Analysis of fluorometric and electrophysiologic data

Images and movies were processed and analyzed with ImageJ (version 1.52p, National Institutes of Health, Bethesda, MD). Movies from PCF experiments were motion-corrected with the "MultiStackReg" ImageJ plugin to compensate for slow drifts of the patch pipette (less than 1 μm/min). The mean intensity of the ROIs was calculated with ImageJ and further analyzed with Igor Pro (version 6.3.7.2, Wavemetrics, Portland, OR). Fluorescence data are presented as ΔF/F. For some experiments, pH signals were calibrated at the end of the recording in order to quantify changes in $pH_i$. To this end, the fluorescence at steady-state after exposure to high-$K^+$/nigericin calibration solution with pH 8.5, 7.5, 6.5, and 5.5 was fitted and used to calculate $pH_i$ (Fig. 3b, c). PCF fluorescence traces from mouse and CHO cells were corrected for slow diffusion of the dye into the cell by subtraction of a mono-exponential fit. Fluorescence traces from ANG-2-loaded sperm were corrected for bleaching by subtraction of a mono-exponential fit. Time constants τ of fluorescent changes were determined by mono-exponential fitting. For complex kinetics, the half-time $t_{1/2}$ was used instead of τ. Current traces from PCF recordings were filtered using a boxcar filter (width 11 data points, i.e., 5.5 ms) in ClampFit (Molecular Devices). Data displayed as box plots represent 25%, 50% (median), and 75% quartiles, with whiskers displaying minimum and maximum data points, except for outliers outside of 1.5-times interquartile range (IQR). Statistical analysis was performed with Prism 9 (version 9.5.1, Graph Pad, Boston, MA). Data sets were tested for normal distribution using the D'Agostino & Pearson test. Based on the results, the following statistical test were chosen: statistical significance was tested with the Mann-Whitney test for data in Figs. 3j and 4d. For data of Fig. 4g, the Kruskal–Wallis test followed by a post hoc Dunn's multiple comparison was performed. For data sets of Figs. 4c, h, 5h, and Supplementary Fig. 5b, a two-way ANOVA followed by a post hoc Tukey's multiple comparison was performed. To test for a difference in pH change in absence and presence of amiloride, the paired $t$-test (Fig. 3e) or the paired Mann–Whitney test (Fig. 3i) were used. Significance was marked in figures with asterisks: * for $p < 0.05$, ** for $p < 0.01$, *** for $p < 0.001$.

Changes in $pH_i$ were used to calculate the changes in $H^+$ by the following equation $\Delta H^+/\Delta pH = -2.3\beta$, wherein ß is the intrinsic buffer capacity. The ß of cells ranges between 40–80 mM[105–107]. For mammalian sperm, β was estimated to range between 30–87 mM[108,109].

### Reporting summary

Further information on research design is available in the Nature Portfolio Reporting Summary linked to this article.

## Data availability

The mass spectrometry proteomics data (Supplementary Data 1 and 2) has been deposited in the ProteomeXchange Consortium via the PRIDE partner repository with the dataset identifier PXD036819. Additionally, the Skyline analysis files have been deposited inPanorama (Data License: CC BY 4.0 und doi: 10.6069/6m29-at05). The UniProt complete *Homo sapiens* proteome sequence data base was used. All other data is available in the article or the source data that is provided as a Source Data file with this paper. All other data are available on request. Source data are provided with this paper.

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

## Acknowledgements

We thank Heike Krause for preparing the manuscript, Dr. R. Pascal (MPINB—caesar) for art work, Dr. D. Wachten (University of Bonn) for providing SLC9C1$^{-/-}$ mice (RRID:MGI:3531312), and J. Meyer im Hagen for preparation of sperm samples. We thank Monika Raabe for help in MS analysis, and Dr. Kuan-Ting Pan and Dr. Christof Lenz for discussion of PRM data acquisition and analysis. H.U. is funded by the Deutsche Forschungsgemeinschaft (SFB1286 and SFB1565, Projektnummer 469281184).

## Author contributions

U.B.K., T.K.B. and H.U. designed the project. E.G. performed all single-cell experiments by fluorescence microscopy and patch-clamp fluorometry. M.A. performed some single-cell imaging measurements. L.M.W., M.N., S.V.K. and H.U. designed, executed, and analyzed MS experiments. U.B.K., T.K.B., and H.U. wrote the manuscript. All authors read and corrected the manuscript.

## Funding

## Competing interests

The authors declare no competing interests.
