## [Peer Review File · Nature Communications]

Control of intracellular pH and bicarbonate by CO₂ diffusion into human spermREVIEWER COMMENTS

Reviewer #1 (Remarks to the Author):

General comments

In this manuscript, Grahn et al report an interesting and important study challenging several existing concepts on the signaling players in the capacitation-associated intracellular alkalization in human sperm, based on the quantitative mass spectrometry and kinetic patch-clamp fluorometry. Different from previously suggested, first, they found the pH_i is set by an amiloride-sensitive Na^+/H^+ exchanger of the SLC9B, but not another type of Na^+/H^+ exchanger SLC9C1 (sNHE) nor HCO_3^- transporters. Second, they also claim that rapid CO_2 diffusion into cells and carbonic anhydrase that readjusts the $CO_2/HCO_3^-/H^+$ equilibrium weigh over HCO_3^- -transport. Finally, they propose that the proton channel Hv1 is not the major driver in alkalizing sperm during capacitation but protect sperm from being acidified by supporting transient increase of pH_i .

However, the reviewer wishes the craftsmanship of the manuscript could have been better to deliver the significance of the study better. As currently stands, this manuscript has problems to be accepted in Nature Communication without minor experimental revision (such as more controls) but certainly extensive text revision. It is not written reader-friendly, even to those in the field. It is necessary to provide introduction, motivation of the study and the relevance, not only the explanation and conclusion of each experiment as so many molecules are scrutinized. Labeling of the figures is particularly not clear – there are even multiple mislabeling which adds confusion - making it very hard to digest the basic information without legend. Currently, a lot of information is being saved for discussion which this reviewer feels useful to explain the background and justification of each experiment.

Major points

1. Motivation of the study: it will be great to state clearly the motivation of the study upfront in Introduction and save the last paragraph to summarize the new findings. This applies to the same for the first section of Results. How was this study conceived while there are already four existing human sperm proteomes even including one that has a similar size as that of the current study? Please highlight the quantitative nature of this advance mass spec analysis in sperm biology where there is no active transcription and translation and absolute (almost) quantification is possible.
2. It is stated that the MS samples are from only three healthy donors. Yet, more information is needed such as proven fertility, age and ethnicity of the donors, especially considering a relative low overlap value of 63%. Is this sufficient overlap/normally expected in the field?
3. It is not clear whether "negative list" are proteins that have never been detected in either one of these three donors? Or they are excluded from the 63% overlap? It also should make it clear whether the proteins in this negative list has ever been identified by other studies.
4. In Figure 2D-E, why are there 4 biological replicates, when only three people were recruited? How were the biological replicates achieved?
5. If the CatSper subunits and solute carrier (the reference is missing for the latter, line 225) copy numbers are similar in this experiment (human) and sea urchin (literature), what is known about published data of CA2 in other species and Slo3? is this consistent with what you see here?
6. In Figure 3, what would happen if NMDG or Amiloride is added to ES+ NH_4Cl buffer without removing NH_4Cl . If NMDG or Amiloride also inhibit the alkalization in ES+ NH_4Cl , this observation would more directly support the author's conclusion that "Line 234: Alkalization is controlled by amiloride-sensitive Na^+/H^+ exchange...." than showing preventing recovery from the acid load?
7. Figure 3 has many issues. It hard to follow the logic. It seems to suggest that SLC9B1 is behind observations from 3E (as it's the only one sensitive to amiloride that is detected in mass spec experiments, and 3E shows amiloride sensitivity). Then why hyperpolarization is tested, and why subsequent experiments are looking back to sNHEs. Summary from Figure 3 experiments is that sNHE/SLC9C does not sustain voltage-gated Na^+/H^+ exchange in human sperm, but again, it is

confusing why sNHE is still a candidate when it seems to be removed as one based on data from 3E. Line 299 – references both 3I and 3J, but 3J does not show abolishment of alkalization by amiloride, unless 3J is incorrectly labelled

8. I think the authors should tone down the saying that “Line 282: In summary, sNHE/SLC9C does not sustain voltage-gated Na^+/H^+ ” exchange in human sperm.” because they might have different activation condition from that of sea urchin sperm and the proteins were detected by mass spec. What is the author’s speculation that SLC9C1/2 would do in human sperm then?

9. This reviewer suggests the author add another group in Figure 3 I, which is ES+NMGD, if the pH_i increasing is totally blocked, that will further the author’s conclusion in “Line 304: amiloride-sensitive Na^+/H^+ exchange rather than sNHE or Hv1 alkalize human sperm during”

10. For this reviewer the content spanning lines 285-305 belongs to the previous subsection, “Alkalization is controlled by amiloride-sensitive ..”, as it is more related and supportive what’s shown previously than what’s coming next.

11. In Figure 4B-D, the author concluded the presence of HCO_3^- does not significantly alter intracellular pH, based on 30mM HCO_3^- being added to the buffer in which pH has already been adjusted to 7.5. What would happen if you do not change pH and HCO_3^- at the same time but one at a time such as adding 30mM HCO_3^- to the pH 6.5 buffer?

12.

Minor points

1. Lines 54 -59 requires citations.

2. line 335, when the author could not exclude HCO_3^- influx via SLC26A3 and SLC26A8, what brought to the conclusion “the rapid acidification demonstrates that CO_2 diffusion across membranes is more significant than HCO_3^- influx”? In the quasi-closed system, the pCO_2 in testis and oviduct is the same 40 mmHg while intracellular alkalization occur during capacitation (oviduct), not in the test. Why does CO_2 diffusion does not affect low intracellular pH in the sperm maintained in the testis? Please discuss

3. Define pH_i = intracellular pH and pH_o =extracellular pH early on when first mentioned in the text.

4. Due to so many proteins are involved in this manuscript, whenever some functions are related, please give the reader some explanation about their function, for example, line 252, ENaC, sodium channel; NHE1, sodium/proton exchanger.

13. Was the calibration done once (Figure 3B?) or is this the same for all of the replicates in 3C? Figure 3I \diamond what is number of experiment done? N-values reported in text (Lines 297, 298) and in figure legends for 3J are not consistent

5. Figure 4 B-D, the difference in pH 6.5 and pH 7.7 is only 3mM HCO_3^- vs 30mM HCO_3^- but both conditions are under 5% CO_2 , then the author should not say “Line 313 in the presence or absence of $\text{CO}_2/\text{HCO}_3^-$. The figure legend also contains the information incorrectly.

6. Figure 4 B - it says pH is 7.5 or 7.7, figure legend says pH_o 7.7, and text referring to figure (311) says it’s 7.5, so, it is confusing what the pH really is. Furthermore, unclear as to the justification of using pH of 7.5 if the goal is to simulate oviduct environment which according to Figure 4A is pH 8.

7. In Figure 4 E, the color description of each trace line in figure legend and figure is conflict. Please make that point clear.

8. In Figure 4E, what does the Alexa488 represent and what is it? It is not clear. What is the justification of using pH 7.35 for experiment in 4E?

9. Experiment in figure 5I is the only one where V_m pulse is +60 mV instead of +70 mV, but no justification on this is given.

Reviewer #2 (Remarks to the Author):

The manuscript describes a multidisciplinary LC-MS/MS, cellular physiology approach to define pH regulatory mechanisms that underlie sperm capacitation in humans. The results suggest that many studies performed in mice do not hold relevance to human sperm, and that even some functional human studies are errant in their assumptions of what proteins are expressed in human sperm. I must

take the LC-MS/MS results at face value. The authors show evidence of roles for Hv1 and an NHE, but stop short of demonstrating their importance or, in the case of the NHE, molecular identity. These aspects weaken the study. The statements in the abstract that "pHi is set by...SLC9B type" and "Hv1...blunts ensuing acidification" do not appear to be adequately demonstrated in this work.

MAJOR:

[1] None of the comparative statements about the data are backed up by statistical analysis.

[2] Although the authors have robustly demonstrated that certain proteins cannot be readily detected by LC-MS/MS in sperm, this is not the same as demonstrating their absence or their functional irrelevance by independent means. Given the iconoclastic nature and likely controversy of these findings, the authors really ought to allocate more discussion to how and why they think previous reports were misleading. This may include replicating some earlier experiments.

[3a] The authors show the importance of an amiloride sensitive NHE, suggesting that it could be SLC9B1 or B2. Yet human SLC9B2 is amiloride insensitive, while the sensitivity of human SLC9B1 is unknown (line 257-258). Presumably then, the protein responsible is not SLC9B2. Typically, the assignment of a cell function to a protein requires testing for demonstrated features of the isolated protein. The authors ought to demonstrate that SLC9B1 is amiloride sensitive in a heterologous system to remove this uncertainty. It also appears unusual that an NHE could alkalinize a cell in response to extracellular alkalinization alone while not performing Na⁺-driven H⁺ efflux during acidification (like NHEs of the SLC9A family do). Has this action been previously described for SLC9B1? If not, again, I suggest the authors turn to a heterologous system to characterize this critical feature of SLC9B1 activity.

[3b] Line 352: There is no evidence for pH recovery in Figure 4 or S4.

[4] Line 544, 560: The authors don't seem to have demonstrated the importance of Hv1 in human sperm, just its activity. Presumably Zn²⁺ would enhance the rate of acidification? As this is one of the major conclusions of the work, a demonstration that Hv1 makes a difference seems to be critical.

[5] The sequence of events in capacitation in this new model, are not clear. The authors seem to be invoking a sudden influx of CO₂ to raise HCO₃⁻, from which there is no recovery save a possible contribution from Hv1, with NHE setting a resting pH. To my understanding, an equally likely model is one in which uterine HCO₃⁻ secretion raises extracellular pH, while generating CO₂, simultaneously activating SLC9B so that CO₂ influx into the sperm, coupled with H⁺ efflux results in a rise in the requisite rise in [HCO₃⁻]_i. The role of Hv1 seems less tangible.

MINOR:

Line 70: Many signaling proteins [in sperm]

Line 76: CA4 is an extracellular enzyme

Line 79 and elsewhere: Authors refer to SLC4 and SLC26A proteins. Standardize as SLC4 and SLC26.

Line 84: and [with] the epithelial sodium channel

Line 96/97: the properties [and cell inventory] of signaling molecules [in] sperm

Line 165: robustness might be a more accurate description than sensitivity and reliability.

Line 247: ES medium. Suggest specific note of lack of HCO₃⁻.

Line 252, 256, 260: NHE1 (SLC9A1)

Line 259, 260: B1 and B2?

Line 247: note that the sNHE-CHO was also sea urchin NHE

Line 277: define V_{1/2}?

Line 301: The direction of change of [Na] ought to be indicated here and in the figure. In general the figures could be better annotated so the reader is less reliant on the legends.

Line 347: I would not describe CA2 as a signaling protein

Line 352: There was no evidence of a pH recovery

Line 361: [either] human [or] mouse sperm

Line 374: Can the authors be confident that these larger currents are also Hv1? Are they Zn blocked?

Line 412: [human] sperm

Line 470: It cannot be concluded that they do not exist.

Line 472: This statement is incorrect. Some SLC4 disruptions cause fertility phenotypes (SLC4A2-null males are sterile), while others have not been tested (SLC4A4-null mice do not survive to breeding age). However, these proteins are expressed elsewhere in the reproductive tract so that doesn't

undermine the authors intended point.

Reviewer #3 (Remarks to the Author):

Review:

Control of intracellular pH and bicarbonate by CO₂ diffusion into human sperm

Summary: In this study, Grahn et al., utilized state-of-the art LC-MS/MS and other methodologies to identify signaling proteins in the mammalian sperm proteome with particular focus on the regulation of intracellular pH and the downstream molecular players involved in the key events that occur during sperm activation. The authors recognize the sizeable gap in our overall knowledge of sperm signaling proteins and further recognize the importance of understanding these systems in the context of the overall proteome. The work substantially adds to a burgeoning field of "reproductive proteomics" something near and dear to this reviewer, and will therefore be of wide interest to cell, developmental and systems biologists. The authors rightly point out the complexity of the system and the current state of the field and highlight the confusion and conundrums currently present. From their studies the authors conclude the need, "... for a substantial revision of signaling concepts in human sperm...". On many levels this reviewer agrees wholeheartedly.

Critique:

The work provides arguably the deepest coverage of a sperm proteome (note that comparably deep *Drosophila* sperm proteomes have also been recently reported- *Mol Cell Proteomics* 21, 100281 (2022) and *Proc National Acad Sci* 119, e2119899119 (2022)).

However, the work's most important contribution lies in the tour-de-force quantitative demonstration (by recognized world class proteomicists) of not only the presence, but also the absence of a suite of signaling proteins involved primarily in the regulation of intracellular pH essential to sperm function. Based on this new quantitative knowledge of the "signaling proteome" the authors then embarked on an equally impressive study and identification of the key players in regulating the all-important CO₂/HCO₃⁻/H⁺ equilibria by the use pharmacological agents and patch-clamping. These studies culminated in a new signaling pathway model for the control of intracellular pH. (Minor point: For the visually challenged, the ? in Figure 6 is barely visible to this reviewers eye, could change to more contrasting color?)

Overall, the work is clearly meritorious for publication, it is well written and the complexity of the downstream signaling pathways following bicarbonate influx and concomitant increase in intracellular pH are clearly presented and should be accessible to a wide readership. A careful reading of the methods section revealed no red flags.

I have only a few minor comments:

General-

1. The authors should include as Supplementary data, excel files of all proteins identified. Unless I missed it, they provide only a statement that the data has been deposited in PRIDE, which is primarily a data storage repository for mass spectra data and reviewers should not have to access this external server to view protein lists.
2. Throughout the narrative, the authors often refer to "sperm", however, as the study is focused only on two species of mammalian sperm, this should be made clearer.
3. Line 106- The sentence beginning, "Our present knowledge about the presence of proteins in human sperm..." is technically incorrect and they probably meant to say, "... signaling proteins in human sperm..." since there are now a number of deep sperm proteomes published including human

and other mammals.

4. Authors explain in Intro (line 109) that drugs studies are "notoriously unreliable..." but then starting on line 244 they report the use of the drug, Nigericin, to study alkalization? They then employed additional pharmacological agents to support the idea that Na/H exchange was the predominant mechanism and further identified, by subtraction of some, and addition of others, that "...SLC9B1, SLC9B2, and SLC9C1/2 are the remaining candidates for Na⁺/H⁺ exchange". Although perhaps a "picky" point, the rhetoric surrounding the use of drugs to study signaling could be perhaps ameliorated.

5. Regarding the proteome depth of coverage, and related to the General comment above, did the final list of all protein IDs include single-hit peptide identifications? I may have missed this, but did not see a specific mention of this important point. Generally 2-peptides are required (as in the above mentioned Drosophila studies) and this would impact any accurate comparisons between proteome ID numbers.

6. The narrative is too long, perhaps a "Results and Discussion" would shorten the overall narrative. Sections of the Discussion are quite repetitive for the most part and combining the two may help to streamline the message and focus on the important points to greater extent.

Specific-

7. The sentence beginning line 422 is a bit ambiguous and could perhaps be written more clearly. "Thus, SLC26A3 would be the dominant if not only HCO₃⁻ transporter in human sperm." There may be a word missing?

8. The extensive protein extraction protocol described was a bit of a puzzle. There was mention of the need to perform a second centrifugation if "a small pellet was observed". How often did this happen? Was it random or did all the extractions need a second centrifugation step? And why not adjust dilution conditions to eliminate this "problem"? This approach is open to criticism if this sample preparation protocol and the resulting insoluble pellet was somehow indicative of a "failed extraction" due to variation in methods/personnel/conditions in which the extractions were performed.

9. Figure 1 legend, no mention of NCKX Na⁺/Ca²⁺ transporter?

Reviewer #1**Major points**

“However, the reviewer wishes the craftsmanship of the manuscript could have been better to deliver the significance of the study better. As currently stands, this manuscript has problems to be accepted in Nature Communication without minor experimental revision (such as more controls) but certainly extensive text revision. It is not written reader-friendly, even to those in the field. It is necessary to provide introduction, motivation of the study and the relevance, not only the explanation and conclusion of each experiment as so many molecules are scrutinized.”

Reading the manuscript indeed requires utmost attention and, in this sense, may not be “reader-friendly”. We believe this is caused by the complexity of the subject and our intention to openly lay out the controversies of this research field rather than by simply poor writing. In fact, reviewer #3 states that the manuscript is well written and adequately addresses the complexity of the field. However, reviewer #3 also noted that the “narrative is too long” and suggests combining Results & Discussion sections. We have accordingly reorganized the manuscript along these lines and layout the rational of each experiment. We hope that this rearrangement renders the manuscript more reader-friendly and addresses also this reviewer’s concerns.

“Labeling of the figures is particularly not clear – there are even multiple mislabeling which adds confusion - making it very hard to digest the basic information without legend.”

We added more information to the figures themselves to facilitate reading without the need resorting to the legend and rectified mislabeling.

“Currently, a lot of information is being saved for discussion which this reviewer feels useful to explain the background and justification of each experiment.”

We followed the suggestion of Reviewer #3 combining Results & Discussion sections and hope that the new arrangement will improve this difficulty and also address the related concerns of reviewer #1.

“Motivation of the study: it will be great to state clearly the motivation of the study upfront in Introduction and save the last paragraph to summarize the new findings. This applies to the same for the first section of Results.”

We have changed the introduction accordingly to the reviewer’s suggestion (lines 123-134).

“It is stated that the MS samples are from only three healthy donors. Yet, more information is needed such as proven fertility, age and ethnicity of the donors, especially considering a relative low overlap value of 63%. Is this sufficient overlap/normally expected in the field?”

We have added the age and ethnicity of the donors. Although the fertility is not known in terms of children, no infertility is known for any of the donors. (lines 1076-1079)

We have changed the text accordingly. It now reads: “In three sperm samples, we detected approximately 5,000 proteins on average, similar to the (Wang *et al.*, 2013) proteome but approximately 4-5-fold more proteins than in the other studies (Supplementary Table S1). A similar deep proteome has been recently reported for *Drosophila* sperm (Garlovsky *et al.*, 2022; McCullough *et al.*, 2022). Of all identified proteins in our study, 63% were detected in all three sperm samples (Figure 2A). Wang *et al.* (2013) reported an overlap of 82% of proteins identified in their analysis; of

note, each of their replicates consists of sperm samples mixed from several donors, whereas we have analyzed proteomes from single donors.” (lines 144-151)

“It is not clear whether “negative list” are proteins that have never been detected in either one of these three donors? Or they are excluded from the 63% overlap? It also should make it clear whether the proteins in this negative list have ever been identified by other studies.”

The negative list includes proteins that never have been detected in any samples from our donors. In previous studies, the presence of these proteins has been inferred from immunoblots and from pharmacological or physiological experiments – otherwise they wouldn't be listed in the negative list. The most comprehensive proteome comparable to our proteome is that by Wang et al. 2013. With a single exception, the proteins in our negative list were also not detected by Wang et al. Although this information was already provided in Table 3 of the original submission, we highlight it now in the text. (lines 186-196)

“In Figure 2D-E, why are there 4 biological replicates, when only three people were recruited? How were the biological replicates achieved?”

This work encompasses many MS experiments beyond the single proteome. For example, the LOD measurements and copy number experiments required many more sperm samples and donors. Thus, the data were obtained from more than three donors and different samples from one and the same donor.

“If the CatSper subunits and solute carrier (the reference is missing for the latter, line 225) copy numbers are similar in this experiment (human) and sea urchin (literature), what is known about published data of CA4 in other species and Slo3? is this consistent with what you see here?”

The reference for SLC9C1 is Trötschel et al. 2020. It was mentioned one line up (223) in the original manuscript. We insert the reference now two times. To our knowledge, copy numbers in sperm have been only determined here (human) and by Trötschel et al. 2020 (sea urchin). We are also not aware of techniques other than MS that were used to estimate copy numbers reliably. So we cannot compare our quantitative results with those of other studies. We have not addressed CA density in sea urchin in Trötschel et al. (2020), simply because $p\text{CO}_2$ and, thereby, HCO_3^- concentrations are low in marine habitats and HCO_3^- isn't directly involved in the specific chemotaxis signaling pathway of marine invertebrates. Thus, a comparison with sea urchin sperm does not lead to more insight, would be even misleading here. However, CA2 and Slo3 have been reported to exist in other species, e.g. mouse. Indeed, Wandernoth et al. (2010) show that CO_2 superfusion and the external, membrane-associated CAIV in human and mice are important for cAMP synthesis and subsequent enhancement of the flagellar beat. We discuss now their results that fully support our conclusion that CO_2 diffusion into sperm is an important source of HCO_3^- . (lines 597-599)

“In Figure 3, what would happen if NMDG or Amiloride is added to ES+NH4Cl buffer without removing NH4Cl. If NMDG or Amiloride also inhibit the alkalization in ES+NH4Cl, this observation would more directly support the author's conclusion that “Line 234: Alkalization is controlled by amiloride-sensitive Na^+/H^+ exchange.....” than showing preventing recovery from the acid load?”

We thank the reviewer for this comment, because it points out potential sources for confusion. In our opinion, such an experiment wouldn't make much sense. A Na^+/H^+ exchanger usually works to export H^+ (because of the inward Na^+ gradient) after acidification. The acid-load experiment has been used for > 30 years to probe Na^+/H^+ exchange activity in many cells. It is robust, reliable, and the underlying rationale is logical. Moreover, NH_4Cl alkalizes sperm by a different, potent mechanisms that cannot be

prevented by a Na^+/H^+ exchange blocker. NH_3 diffuses – like CO_2 – readily across membranes and the $\text{NH}_3 + \text{H}^+ = \text{NH}_4^+$ equilibrium is re-established inside the cell.

However, we noticed that we use the word ‘alkalization’ for two different processes, which may cause confusion: alkalization starting from the resting state, and recovery from acidification, i.e., removal of H^+ from the cell; although the pH_i increases, it does not represent an alkalization in a pure sense. The cytosol just becomes less acidic. Accordingly, we have changed the subheading into “Recovery from acidification is controlled.....” (line 350)

“Figure 3 has many issues. It hard to follow the logic. It seems to suggest that SLC9B1 is behind observations from 3E (as it’s the only one sensitive to amiloride that is detected in mass spec experiments, and 3E shows amiloride sensitivity). Then why hyperpolarization is tested, and why subsequent experiments are looking back to sNHEs. Summary from Figure 3 experiments is that sNHE/SLC9C does not sustain voltage-gated Na^+/H^+ exchange in human sperm, but again, it is confusing why sNHE is still a candidate when it seems to be removed as one based on data from 3E. Line 299 – references both 3I and 3J, but 3J does not show abolishment of alkalization by amiloride, unless 3J is incorrectly labelled.”

As mentioned here and in other papers (e.g.; Kaupp & Strünker 2017) homologous molecules in different sperm species feature different physiological and pharmacological properties that serve different functions. The sNHE/SLC9C1 in sea urchin indeed is not amiloride-sensitive. However, the human sNHE might be amiloride-sensitive considering that we also detected SLC9C2 and these proteins may form hetero-dimers that are amiloride-sensitive. Thus, data shown in Fig. 3E do not exclude sNHE as Na^+/H^+ exchanger *per se*. In fact, what we show is that the sNHE does *not* mediate *voltage-gated* and *cAMP-modulated* Na^+/H^+ exchange like the sea urchin homologue. This conclusion is also supported by sequence analysis: the transport domain, the S4 voltage-sensor motif, and the CNBD are lacking important functional residues (see Windler et al. 2018). Therefore, the human/mammalian SLC9C1 may not even serve as a voltage-gated Na^+/H^+ exchanger. We now emphasize that we cannot exclude that SLC9C1 might serve as an exchanger, which however - unlike its sea urchin homologue – is not gated by voltage and cAMP and may have acquired amiloride sensitivity either on its own or by forming heterodimers with SLC9C2. (lines 410-414)

The sentence line 299 is correct and labeling of Fig. 3I, J is also correct. To prevent any misunderstanding, we refer now to the figures separately for Zn^{2+} and amiloride: Fig. 3I, J for Zn^{2+} and only Fig. 3I for amiloride. (lines 453-454)

“I think the authors should tone down the saying that “Line 282: In summary, sNHE/SLC9C does not sustain voltage-gated Na^+/H^+ ” exchange in human sperm.” because they might have different activation condition from that of sea urchin sperm and the proteins were detected by mass spec. What is the author’s speculation that SLC9C1/2 would do in human sperm then?”

This comment is related to the previous comment. The sentence itself is correct but the reviewer argues that it is incomplete and requires some amendment or clarification. We agree. Due to the fact that properties of sperm proteins differ markedly among species, we cannot discard the possibility the sNHE may be able to exchange Na^+ for H^+ . Accordingly, we suggest now that sNHE may have lost its voltage/cAMP-dependence and in combination with SLC9C2 may indeed promote Na^+/H^+ exchange. See also our answer to the previous question. We and others have tried to heterologously express the mammalian SLC9C1 – without success. The same holds for SLC9B1. Once the transport mode and substrate specificity of these SLCs are rigorously established, the contribution of each molecule to Na^+/H^+ exchange can be judged.

“This reviewer suggests the author add another group in Figure 3 I, which is ES+NMGD, if the pH_i increasing is totally blocked, that will further the author’s conclusion in “Line 304: amiloride-sensitive Na^+/H^+ exchange rather than sNHE or Hv1 alkalize human sperm during”

As mentioned in the previous two answers, there is uncertainty about pharmacology and transported substrate of the sNHE transporter. We also show in Figure 3E that exchange with NMDG is abolished.

“For this reviewer the content spanning lines 285-305 belongs to the previous subsection, “Alkalization is controlled by amiloride-sensitive”, as it is more related and supportive what’s shown previously than what’s coming next.”

As suggested, we moved the paragraph to the previous subsection.

“In Figure 4B-D, the author concluded the presence of HCO_3^- does not significantly alter intracellular pH , based on 30mM HCO_3^- being added to the buffer in which pH has already been adjusted to 7.5. What would happen if you do not change pH and HCO_3^- at the same time but one at a time such as adding 30mM HCO_3^- to the $pH 6.5$ buffer?”

We do not conclude that the presence of HCO_3^- does not significantly alter pH_i . In fact, we state that the change in pH_i upon a switch from $pH 6.5$ to 7.7 is similar with and without HCO_3^- .

Minor Points

“Lines 54 -59 requires citations.”

We inserted several citations for reviews. (lines 59-60)

“line 335, when the author could not exclude HCO_3^- influx via SLC26A3 and SLC26A8, what brought to the conclusion “the rapid acidification demonstrates that CO_2 diffusion across membranes is more significant than HCO_3^- influx”? In the quasi-closed system, the pCO_2 in testis and oviduct is the same 40 mmHg while intracellular alkalization occur during capacitation (oviduct), not in the test. Why does CO_2 diffusion does not affect low intracellular pH in the sperm maintained in the testis? Please discuss”

HCO_3^- influx via SLCs causes alkalization, whereas CO_2 influx causes acidification. In the end the net change in pH_i , be it acidification or alkalization, depends on the relative contributions of the two mechanisms. Because addition of CO_2/HCO_3^- causes acidification, CO_2 influx must prevail over HCO_3^- influx in mature sperm. In the epididymis, an acidic luminal milieu is maintained by several mechanisms. Principal and clear cells promote reabsorption of HCO_3^- and proton-secretion respectively. Therefore, the bicarbonate concentration is low.

“Define pH_i = intracellular pH and pH_o =extracellular pH early on when first mentioned in the text.”

We have followed the reviewer’s advice. (line 69)

“Due to so many proteins are involved in this manuscript, whenever some functions are related, please give the reader some explanation about their function, for example, line 252, ENaC, sodium channel; NHE1, sodium/proton exchanger.”

There is a fine line or balance between repetitive words (redundancy) and readability. We scrutinized the text and tried to navigate this balance in the new version.

“Was the calibration done once (Figure 3B?) or is this the same for all of the replicates in 3C? Figure 3I ∅ what is number of experiment done? N-values reported in text (Lines 297, 298) and in figure legends for 3J are not consistent”

The calibration was done for every single cell; a representative example is shown in Fig. 3B. Fig. 3C shows the summarized calibrations for 16 cells of 6 experiments. We corrected the N-values.

“Figure4 B-D, the difference in pH6.5 and pH7.7 is only 3mM HCO₃⁻ vs 30mM HCO₃⁻ but both conditions are under 5%CO₂, then the author should not say “Line 313 in the presence or absence of CO₂/HCO₃⁻. The figure legend also contains the information incorrectly.”

The legend was corrected.

“Figure 4 B - it says pH is 7.5 or 7.7, figure legend says pH_o 7.7, and text referring to figure (311) says it's 7.5, so, it is confusing what the pH really is. Furthermore, unclear as to the justification of using pH of 7.5 if the goal is to simulate oviduct environment which according to Figure 4A is pH 8.

In Fig. 4B, the black line shows the pH_i change upon switching from pH_o 6.5 to pH_o 7.7 (no CO₂/HCO₃⁻, same recording as in Fig. 2I). The blue line shows the pH_i change upon switching from 6.5 (5%/3 mM HCO₃⁻) to 7.5 (5% CO₂/30 mM HCO₃⁻). To avoid this confusion, we introduced now two bars in Fig.4B, one with a change of pH alone (black) and one with HCO₃⁻ present (blue). The legend was changed accordingly.

According to the literature, the pH measured in the mammalian oviduct environment varies substantially between 7.1 and 8.2 (Vishvakarma 1962, rabbit), 7.28-7.7 (David et al. 1973, human), 7.1-7.8 (Maas et al. 1977 Rhesus Monkey). A summary can be found in Ng et al. 1918. To reflect this broad range, we changed the pH value in the cartoon for the oviduct environment (Figure 4A, right) to pH>7.4.

“In Figure 4 E, the color description of each trace line in figure legend and figure is conflict. Please make that point clear.”

The legend was corrected.

“In Figure 4E, what does the Alexa488 represent and what is it? It is not clear. What is the justification of using pH 7.35 for experiment in 4E?”

Alexa488 is an inert fluorescent dye that was used to resolve the mixing time. It is described in the Fig. 4E legend of the manuscript. pH 7.35 was chosen to use one and the same pH for both sperm and the two cell lines.

“Experiment in figure 5I is the only one where V_m pulse is +60 mV instead of +70 mV, but no justification on this is given.”

We experienced that more positive and prolonged voltage steps, e.g. to +70 mV and above, often broke the recording of the CHO-Hv1 cell line. However, the CHO-Hv1 cell line strongly overexpresses hHv1; thus a smaller voltage step to +60 mV was chosen and sufficient to get a sizable proton current and subsequent intracellular acidification.

Reviewer #2

“The results suggest that many studies performed in mice do not hold relevance to human sperm...”

The manuscript doesn't state this. However, we emphasize that signaling in mice vs. human sperm is different and we caution against rash interpretation of experiments on mouse sperm with respect to human sperm.

“that even some functional human studies are errant in their assumptions of what proteins are expressed in human sperm. I must take the LC-MS/MS results at face value. The authors show evidence of roles for Hv1 and an NHE, but stop short of demonstrating their importance or, in the case of the NHE, molecular identity. These aspects weaken the study. The statements in the abstract that “pHi is set by...SLC9B type” and “Hv1...blunts ensuing acidification” do not appear to be adequately demonstrated in this work.”

This study amply shows how Hv1 responds to changes in pHi not only in sperm but also cell lines, in particular acidification by state-of-the-art patch-clamp fluorimetry that – to our knowledge – has not yet been used before to measure changes in pHi or [Ca²⁺]. Thereby, we could show that sNHE does not – as this research field expected – mediate *voltage gated* and *cAMP-sensitive* Na⁺/H⁺ exchange. We have rephrased the statements in the Abstract and laid out the arguments in the discussion paragraphs the pros and cons with respect to sNHE and SLC9B1 vs. SLC9B2.

“None of the comparative statements about the data are backed up by statistical analysis. “

We thank the reviewer for pointing this out. Accordingly, we provide this analysis now.

“Although the authors have robustly demonstrated that certain proteins cannot be readily detected by LC-MS/MS in sperm, this is not the same as demonstrating their absence or their functional irrelevance by independent means. Given the iconoclastic nature and likely controversy of these findings, the authors really ought to allocate more discussion to how and why they think previous reports were misleading. This may include replicating some earlier experiments.”

We understand that the reviewer used the word “iconoclastic” in a figurative sense to indicate how consequential, disturbing, and difficult to digest our findings are. Indeed, this work seriously questions several decades of fertilization research. The mantra, not only in sperm research, has been that “if you can't identify a molecule (protein), it doesn't mean it doesn't exist or is irrelevant.” We do away with the arbitrariness of this argument by rigorously quantitative MS (LOD) approach that seriously constrains alternative explanations or eliminates places where one can hide. We discuss why such low copy numbers (< 10), owing to the generally low turnover number of SLCs, are unable to significantly change cytosol concentrations, in particular for [HCO₃⁻] or [Cl⁻]. We also specifically discuss now how this quantification constrains hypotheses about the relative contributions of proteins that serve similar or identical function (e.g., SLC isoforms that transport HCO₃⁻, form hetero-dimers, form oligomers with ion channels, or the contribution of ion channels to V_m or membrane currents).

There are virtually several hundreds of papers that study sperm molecules from our “negative list”. It certainly is beyond the scope of this manuscript to discuss the potential fallacies of previous research, leaving alone repeating it. Most of the conclusions were previously derived using pharmacological and immunological tools. However, one of our crucial findings is that external HCO₃⁻ acidifies rather than alkalizes sperm. We discuss possible explanations why previously, in some reports, an alkalization was measured.

“The authors show the importance of an amiloride sensitive NHE, suggesting that it could be SLC9B1 or B2. Yet human SLC9B2 is amiloride insensitive, while the sensitivity of human SLC9B1 is unknown (line 257-258). Presumably then, the protein responsible is not SLC9B2. Typically, the assignment of a cell function to a protein requires testing for demonstrated features of the isolated protein. The authors ought to demonstrate that SLC9B1 is amiloride sensitive in a heterologous system to remove this uncertainty. It also appears unusual that an NHE could alkalinize a cell in response to extracellular alkalinization alone while not performing Na⁺-driven H⁺ efflux during acidification (like NHEs of the SLC9A family do). Has this action been previously described for SLC9B1? If not, again, I suggest the authors turn to a heterologous system to characterize this critical feature of SLC9B1 activity.”

We agree with the reviewer that SLCB1 and B2 are less characterized than other members of the SLC9 family (see the recent review by Anderegg et al. 2022). However, the work by Balbach et al. 2020 in mouse sperm strongly argues that SLC9B1 serves as a Na⁺/H⁺ exchanger and in SLC9B1^{-/-} sperm, most of the zona pellucida-evoked pH_i changes are abolished. The residual exchange activity might be due to SLC9B2. Similar results were obtained by Chen et al. 2016. Because SLC9B2 is not amiloride-sensitive and serves in subcellular organelles, we think it is unlikely that it mediates Na⁺/H⁺ exchange under our conditions. We have also tried to functionally express SLC9B1 in oocyte – without success.

“Line 352: There is no evidence for pH recovery in Figure 4 or S4.”

We are not sure to understand correctly what the reviewer refers to in line 352 and Figure 4. Furthermore, pH recovery is reported in more detail in Figure 5.

“Line 544, 560: The authors don’t seem to have demonstrated the importance of Hv1 in human sperm, just its activity. Presumably Zn²⁺ would enhance the rate of acidification? As this is one of the major conclusions of the work, a demonstration that Hv1 makes a difference seems to be critical.”

We have clearly shown or argued that (i) owing to its peculiar activation mechanism, H_v1 cannot produce a large alkalization on its own during capacitation. (ii) CO₂/HCO₃⁻ stimulates H_v1 activity not by some mysterious effect of capacitation but by an increase of ΔpH across the membrane and therefore, shifts P_o of H_v1 by a well-documented mechanism and, thereby, activates the channel; (iii) H_v1 cannot maintain a long-lasting effect on pH_i. Once H_v1 closes, adjustment of pH by the CO₂/HCO₃⁻/H⁺ equilibrium kicks in within seconds.

“The sequence of events in capacitation in this new model, are not clear. The authors seem to be invoking a sudden influx of CO₂ to raise HCO₃, from which there is no recovery save a possible contribution from Hv1, with NHE setting a resting pH. To my understanding, an equally likely model is one in which uterine HCO₃ secretion raises extracellular pH, while generating CO₂, simultaneously activating SLC9B so that CO₂ influx into the sperm, coupled with H⁺ efflux results in a rise in the requisite rise in [HCO₃]_i. The role of Hv1 seems less tangible.”

We like the reviewer’s comment very much. It’s food for thought to describe the events even clearer. Our model isn’t much different from his/her alternative suggestion. The reviewer looks at different parts of the equilibrium: his/her model is mainly concerned with mechanisms that change the *extracellular* medium and, consequently, intracellular milieu as well, whereas our model just assumes that the oviductal pH is much more alkaline and the HCO₃⁻ concentration is high. We state in the introduction that the CO₂/HCO₃⁻/H⁺ equilibrium applies for both the extra- and intracellular compartments. The two equilibria are connected via CO₂ diffusion and HCO₃⁻ and H⁺ transport across the membrane not only of sperm but also the surrounding epithelial cells. We agree with all other reports that a rise of intracellular HCO₃⁻ is important. However, we disagree that intracellular alkalization reflects HCO₃⁻ influx. Instead, we propose that continuous CO₂ influx substantially

contributes to intracellular HCO_3^- synthesis. The ensuing acidification – that’s our hypothesis – is ameliorated by H^+ efflux via Hv1, and, thereby, HCO_3^- synthesis is sustained for some time. Otherwise, owing to the acidification and its action on the $\text{CO}_2/\text{HCO}_3^-/\text{H}^+$ equilibrium, synthesis would cease. In our experiments, we impose a sudden CO_2 influx by relatively rapid $\text{CO}_2/\text{HCO}_3^-$ perfusion to study these mechanisms of $\text{HCO}_3^-/\text{pH}_i$ regulation in sperm; in a more physiological setting of the oviduct, sperm are probably exposed to slower changes in extracellular HCO_3^- and pH (which we address experimentally in Figure 4). Finally, from first physico-chemical principles, a lower pH_i is also sensed by most NHE exchanger that use the alternate access mechanism (except for the sea urchin variety of sNHEs that require gating by voltage/cAMP) and, therefore, Na^+/H^+ exchange may also contribute to H^+ extrusion. Because, judging from iBAQ values, Hv1 is one of the most abundant proteins and because channels usually support a much higher flux rate than SLCs, it is plausible that Hv1 dominates H^+ efflux required to ameliorate the acidification produced by HCO_3^- synthesis from CO_2 . An important unknown is the sNHE(SLC9C1/2). We clearly state that the substrate specificity of sNHE is in limbo as well as its function and mode of operation. However, in the reviewer’s spirit that Hv1 seems less tangible we mention in the text that future work needs to address why human sperm need both Na^+/H^+ exchange **and** Hv1 for proton extrusion, whereas mouse sperm is left with only Na^+/H^+ exchange. (lines 626-628)

Minor Points

“Line 70: Many signaling proteins [in sperm]”

As part of the major revision of the manuscript, we changed the sentence to “Many proteins and mechanisms have been implied in the execution and coordination of these three signaling events (Figure 1) (Nishigaki et al., 2014; Puga Molina et al., 2018a)”. From the context, it is now clear that the sentence refers to sperm signaling events (Fig. 1). (line77-78)

“Line 76: CA4 is an extracellular enzyme”

Yes, but is it necessary to highlight this here? For the protocol, CA4 is tethered to the extracellular side of the membrane. Any change of its activity will also affect CO_2 diffusion or HCO_3^- transport across the membrane. The pertinent refs. are given.

“Line 79 and elsewhere: Authors refer to SLC4 and SLC26A proteins. Standardize as SLC4 and SLC26.”

We now refer to family SLC4 and SLC26 in order to stay consistent.

“Line 84: and [with] the epithelial sodium channel”

Changed

“Line 96/97: the properties [and cell inventory] of signaling molecules [in] sperm”

Changed

“Line 165: robustness might be a more accurate description than sensitivity and reliability.”

All three terms convey slightly different yet overlapping aspects. We thank the reviewer for this suggestion and use now all three words. (Line 177-178)

“Line 247: ES medium. Suggest specific note of lack of HCO₃.”

Has been amended. (line 360)

“Line 252, 256, 260: NHE1 (SLC9A1)”

We removed SLC9A, because its not present in sperm.

“Line 259, 260: B1 and B2?”

Text has been removed in the new version.

“Line 247: note that the sNHE-CHO was also sea urchin NHE”

Line is wrong. It is line 274. We amended the text. (lines 389-390)

“Line 277: define V_{1/2}?”

We define now V_{1/2}. (line 393)

“Line 301: The direction of change of [Na] ought to be indicated here and in the figure. In general, the figures could be better annotated so the reader is less reliant on the legends.”

We, in fact, mean ΔpH or $\Delta[\text{Na}]_i$. We indicate the flux direction when necessary. We followed the suggestion of the reviewer and scrutinized all our figures in this respect.

“Line 347: I would not describe CA2 as a signaling protein”

We removed the word “signaling” and phrased it differently. (line 503)

“Line 352: There was no evidence of a pH recovery”

We are not sure which sentence the reviewer is referring to. In line 352, there is no mentioning of pH recovery. Figure 5 shows data on pH recovery.

“Line 361: [either] human [or] mouse sperm”

Changed

“Line 374: Can the authors be confident that these larger currents are also Hv1? Are they Zn blocked?”

We did not test the Zn²⁺ sensitivity of those currents in the presence of HCO₃⁻. However, we are fairly confident that the larger currents are also mediated by Hv1. 1) because pH_i is reduced in CO₂/HCO₃⁻, there is an increased driving force for protons; thus, larger proton currents via Hv1 for the same voltage step are expected. 2) The larger current displays the “drooping” (apparent inactivation, but due to shifting the proton gradient during channel opening [proton depletion and accumulation at intra- end extracellular side of the membrane, respectively]), which is typical for large Hv1-mediated currents

(see e.g. De-la-Rosa et al. J Gen Physiol 2016). 3) There is no other proton-conducting, depolarization-activated channel expressed in human sperm.

“Line 412: [human] sperm”

We added the word „human“. (line 630)

“Line 470: It cannot be concluded that they do not exist.”

This part has been completely rewritten (lines 283-310)

“Line 472: This statement is incorrect. Some SLC4 disruptions cause fertility phenotypes (SLC4A2-null males are sterile), while others have not been tested (SLC4A4-null mice do not survive to breeding age). However, these proteins are expressed elsewhere in the reproductive tract so that doesn’t undermine the authors intended point.”

We phrased this part more specifically. (lines 325-330)

Reviewer #3

“The work provides arguably the deepest coverage of a sperm proteome (note that comparably deep Drosophila sperm proteomes have also been recently reported- Mol Cell Proteomics 21, 100281 (2022) and Proc National Acad Sci 119, e2119899119 (2022)).”

We added these references. (lines 147-148)

“However, the work’s most important contribution lies in the tour-de-force quantitative demonstration (by recognized world class proteomicists) of not only the presence, but also the absence of a suite of signaling proteins involved primarily in the regulation of intracellular pH essential to sperm function. Based on this new quantitative knowledge of the “signaling proteome” the authors then embarked on an equally impressive study and identification of the key players in regulating the all-important CO₂/HCO₃⁻/H⁺ equilibria by the use pharmacological agents and patch-clamping. These studies culminated in a new signaling pathway model for the control of intracellular pH. (Minor point: For the visually challenged, the ? in Figure 6 is barely visible to this reviewers eye, could change to more contrasting color?)”

We scrutinized all our figures for visibility and enhanced contrast accordingly.

“The authors should include as Supplementary data, excel files of all proteins identified. Unless I missed it, they provide only a statement that the data has been deposited in PRIDE, which is primarily a data storage repository for mass spectra data and reviewers should not have to access this external server to view protein lists.”

We have deposited all original data for MS in a data base and have provided this information to the editorial office of Nature Communications.

“Throughout the narrative, the authors often refer to “sperm”, however, as the study is focused only on two species of mammalian sperm, this should be made clearer.”

We followed the advice of this reviewer and accordingly highlighted the sperm species when it seemed necessary.

“Line 106- The sentence beginning, “Our present knowledge about the presence of proteins in human sperm...” is technically incorrect and they probably meant to say, “... signaling proteins in human sperm...” since there are now a number of deep sperm proteomes published including human and other mammals.”

We modified this sentence accordingly. (lines 116-118)

“Authors explain in Intro (line 109) that drugs studies are “notoriously unreliable...” but then starting on line 244 they report the use of the drug, Nigericin, to study alkalization? They then employed additional pharmacological agents to support the idea that Na/H exchange was the predominant mechanism and further identified, by subtraction of some, and addition of others, that “...,SLC9B1, SLC9B2, and SLC9C1/2 are the remaining candidates for Na⁺/H⁺ exchange”. Although perhaps a “picky” point, the rhetoric surrounding the use of drugs to study signaling could be perhaps ameliorated.”

We thank the reviewer for this comment. The point is anything but “picky”; it addresses an important issue faced by the entire research field: how specific are drugs and at what concentrations? In our opinion, some of the issues that this manuscript addresses are probably caused by the use of drugs along with shaky statistics. In fact, this manuscript reminds the reader of how unreliable pharmacology can be. How can a drug produce a significant effect when the “specific” target is absent? While we are on this subject, quantitative MS targeting the lowest copy number (LOD value) might become a great method to reveal unspecific effects of drugs. Notwithstanding, we tone down this issue to more palatable phrasing. Finally, we employed drugs to show that, unlike previous reports, we often didn't see a significant effect. Thus, our pharmacology agrees with the MS results. (lines 118-122)

“Regarding the proteome depth of coverage, and related to the General comment above, did the final list of all protein IDs include single-hit peptide identifications? I may have missed this, but did not see a specific mention of this important point. Generally, 2-peptides are required (as in the above mentioned Drosophila studies) and this would impact any accurate comparisons between proteome ID numbers.”

To answer the reviewer's question, we have also performed a database search with settings described in the Methods section but with two peptides required for protein identification. Of note, for this the MaxQuant version 2.1.4.0 was used and identified 6,097 unique proteins from all three replicates. 6,430 unique proteins have been identified in our previous search from all the replicates and allowing for one peptide (Supplementary Table S2). The overlap between our three replicates is 64% when two peptides are considered for the identification of one protein. Thus, there is no difference in the percentage of overlap of proteins in the three replicates when one or two peptides are allowed for search. Importantly, all proteins mentioned in the manuscript are also identified in the search with two peptides required for identification.

Because of these small differences we would like to keep the database search and the presentation of the results as they are. Since the MS data are submitted to a public repository, everyone can perform a search with settings according to his/her needs.

"The narrative is too long, perhaps a "Results and Discussion" would shorten the overall narrative. Sections of the Discussion are quite repetitive for the most part and combining the two may help to streamline the message and focus on the important points to greater extent."

The reviewer's suggestion is quite sensible. Indeed, the manuscript is hard to digest. The narrative is so long, because the issue is so complex and our conclusions are iconoclastic, as a reviewer dryly noted. Therefore, we intended to describe issues as comprehensive and precise as possible, which apparently compromised the clarity of the text. We have accordingly entirely re-organized the manuscript along these lines: summarize briefly the aim of the study and the results at the end of the introduction, combine Results & Discussions and finish with a terse conclusion.

"The sentence beginning line 422 is a bit ambiguous and could perhaps be written more clearly. "Thus, SLC26A3 would be the dominant if not only HCO₃⁻ transporter in human sperm." There may be a word missing?"

We agree and changed the text around this sentence accordingly (lines 302-310).

"The extensive protein extraction protocol described was a bit of a puzzle. There was mention of the need to perform a second centrifugation if "a small pellet was observed". How often did this happen? Was it random or did all the extractions need a second centrifugation step? And why not adjust dilution conditions to eliminate this "problem"? This approach is open to criticism if this sample preparation protocol and the resulting insoluble pellet was somehow indicative of a "failed extraction" due to variation in methods/personnel/conditions in which the extractions were performed."

We have corrected this. We have changed the Materials and Method section. It now reads: "To control for complete solubilization and protein recovery, human sperm cells were denatured as described above and then in an additional step centrifuged at 30,000xg (26,000 rpm, S100-AT4 rotor, UZ MX150, Thermo Sorvall) to separate insoluble material. The residual small pellet was again dissolved in 8 % (w/v) SDS, digested, and analyzed by LC-MS. Furthermore, to control for lost peptides during enzymatic digestion, SP3 beads were - after the above- described elution steps of peptides – incubated with 50 % (v/v) ACN and 0.1 % (v/v) FA for 15 min at 37 °C. The solution containing eventually residual extracted peptides was dried in a SpeedVac concentrator and then analyzed by LC-MS as above."(lines 1137-1144)

Figure 1 legend, no mention of NCKX Na⁺/Ca²⁺ transporter?

We define now NCKX as Na⁺/K⁺-Ca²⁺ transporter.

REVIEWER COMMENTS

Reviewer #1 (Remarks to the Author):

The revised manuscript of "Control of intracellular pH and bicarbonate by CO₂ diffusion into human sperm" is edited/modified as this reviewer suggested accordingly. Indeed, the text is more reader-friendly and the logic is easier to follow by combining the Results with Discussion. The major points and minor points are addressed by the authors very clearly.

Several minor points.

1. Different what is stated from the author's rebuttal, the age of sperm donors is not given in the revised manuscript (lines 1076-1079) but only ethnicity as it states "healthy adult Caucasian males". It might be prudent to use a specific applicable definition (e.g., 18-55 old).
2. In Fig. 6, as the Ca²⁺ influx mediated by CatSper channel is not primarily from PMCA4 exported Ca²⁺, it would be better to remove the arrow in the extracellular side between PMCA4 and CatSper. For the CatSper channel, EFCAB9 and CATSPERZ form a subcomplex together, it will be more accurate to add zeta together if EFCAB9 to be included in the cartoon. Or the authors could just remove EFCAB9, the left channel could simply represent the complete, functional CatSper channel.
3. Line 1145: Please correct HEK292 to HEK293

Reviewer #2 (Remarks to the Author):

I am satisfied with the authors' responses to my comments and the actions taken to adjust the manuscript. I have a few remaining questions, all related to the newly included statistical analyses:

[1] How did the authors determine that the Kolmogorov-Smirnov test is most appropriate for their statistical analyses rather than the more usual t-test and ANOVA (for multiple comparisons)? This seems like an unusual choice without clarification. It is my understanding that K-S is more appropriate than a t-test in the case of non-normally distributed data. Did the authors determine that their data were not normally distributed? An explanation in the methods would be valuable to the reader.

[2] Figure 4G. For the amiloride sensitivity of the pHi homeostasis P=0.02. It would be expected for the multiple comparisons in 4G that ANOVA be applied or that the P value threshold for t-test/K-S test be adjusted for multiple comparisons, in which case P would need to be <0.01 to be significant. The significance of amiloride sensitivity is of key importance to identifying the SLC9 responsible.

[3] Can the authors define and explain the relationship between nexp and ncells. Does nexp represent different days or different dishes of cells etc.? A more rigorous statistical analysis could include batch dependence as a nested factor: this may help with the analysis of inhibitor effect.

[4] Some statements are still lacking statistical backing, which the authors could easily calculate. For example line 494-496 appears to be based on comparison averages rather than a demonstration of significant difference among cell types in 4H data.

[5] In any case, a two-way ANOVA or equivalent would be more appropriate for Fig 4C, 4H, and 5H data.

Reviewer #3 (Remarks to the Author):

Overall the authors have adequately responded to all comments except one- a final list of all proteins

identified by mass spec should be included in a Supplemental table (preferably excel) that contains details of the protein accession and the relevant quantitative information.

Reviewer #1 (Remarks to the Author):

The revised manuscript of "Control of intracellular pH and bicarbonate by CO₂ diffusion into human sperm" is edited/modified as this reviewer suggested accordingly. Indeed, the text is more reader-friendly and the logic is easier to follow by combining the Results with Discussion. The major points and minor points are addressed by the authors very clearly.

Several minor points.

1. Different what is stated from the author's rebuttal, the age of sperm donors is not given in the revised manuscript (lines 1076-1079) but only ethnicity as it states "healthy adult Caucasian males". It might be prudent to use a specific applicable definition (e.g., 18-55 old).

We provide now the age range of the donors (21-56 y).

2. In Fig. 6, as the Ca²⁺ influx mediated by CatSper channel is not primarily from PMCA4 exported Ca²⁺, it would be better to remove the arrow in the extracellular side between PMCA4 and CatSper. For the CatSper channel, EFCAB9 and CATSPERZ form a subcomplex together, it will be more accurate to add zeta together if EFCAB9 to be included in the cartoon. Or the authors could just remove EFCAB9, the left channel could simply represent the complete, functional CatSper channel.

We changed the cartoon following the reviewer's suggestions: we removed the arrow between PMCA4 and CatSper and the icon for AFCAB9.

3. Line 1145: Please correct HEK292 to HEK293

Corrected.

Reviewer #2 (Remarks to the Author):

I am satisfied with the authors' responses to my comments and the actions taken to adjust the manuscript. I have a few remaining questions, all related to the newly included statistical analyses:

We particularly thank this reviewer for the valuable comments and follow most of his/her suggestions and accordingly re-examined and revised the statistical analysis of our data. Where we partially disagree (point [5]), we explain the reasons but nonetheless follow her/his suggestions.

[1] How did the authors determine that the Kolmogorov-Smirnov test is most appropriate for their statistical analyses rather than the more usual t-test and ANOVA (for multiple comparisons)? This seems like an unusual choice without clarification. It is my understanding that K-S is more appropriate than a t-test in the case of non-normally distributed data. Did the authors determine that their data were not normally distributed? An explanation in the methods would be valuable to the reader.

In the original manuscript, we did not include statistical analysis. For the revised version, we used two different methods to test for the normal distribution of the data sets: the test by D'Agostino & Pearson and the test by Shapiro using the Prism 9 (Graph Pad) software. The results were mixed: for some data (i.e., a certain condition) within a dataset, the hypothesis that the data is normally

distributed was rejected; for other data from the same dataset, the hypothesis was accepted. For example, normal distribution for ES data (control condition) was rejected but accepted for the Zn^{2+} condition in Fig. 3j. For some dataset, there were also differences between the two methods that tested normal distribution. Therefore, we decided to use a non-parametric test that does not require normal distribution of the data. A non-parametric test has usually less power when applied to normally distributed data; i.e., it is less likely to detect a true difference. We preferred this conservative choice over using standard tests that require normality of the data. Arguably, the two-sample Kolmogorov-Smirnov test is a test with high power and, although less frequently used, appropriate for testing differences in the distribution of two groups. Following the suggestions of this reviewer, we re-analyzed our data using different tests that are more commonly used given the dataset structures and sample sizes (see replies below), and now also mention the use of the test for normality in the methods section. Tests were also performed with Prism 9 (Graph Pad).

[2] Figure 4G. For the amiloride sensitivity of the pH_i homeostasis $P=0.02$. It would be expected for the multiple comparisons in 4G that ANOVA be applied or that the P value threshold for t-test/K-S test be adjusted for multiple comparisons, in which case P would need to be <0.01 to be significant. The significance of amiloride sensitivity is of key importance to identifying the SLC9 responsible.

We agree that, for the dataset of Figure 4G, an analysis method for multiple groups is appropriate; we now implemented this test in the new version of the manuscript. Most conditions are not normally distributed (except the $CFTR_{inh}$ condition), therefore we chose the nonparametric Kruskal-Wallis test to identify whether there is a difference between the groups (testing variances using the ranks of the data). Because the dataset contains a control group, we chose a test using multiple comparisons of the various experimental conditions with the control group. Because our dataset does not have equal sample sizes, the method of choice is the Dunn method (as recommended by Zar JH Biostatistical analysis, Pearson, 5th edition, 2009, p. 243). Using this method, no significant difference is detected between any condition and the control group. Therefore, we change the text describing the data in the results section, similar to the wording of the initial version of the manuscript.

The amiloride sensitivity may not be of key importance to identify SLC9 as being responsible for H^+ extrusion. Although we clearly show amiloride-sensitive pH regulation in sperm using the acid-load experiment described in Figure 3D&E and the lack of change in pH_i in the presence of amiloride when perfusing with an alkaline solution (Figure 3I), the amiloride effect may not show under other conditions. We can only speculate why amiloride fails to have a significant effect on CO_2/HCO_3^- -induced acidification in sperm. The activity of SLC9 is likely to be pH_i -dependent; at low pH_i , SLC9 should have higher proton transport rates, which could be the reason why this is easier to detect fluorometrically (as in Fig. 3D,E & I). As we describe in the text, to answer this issue unequivocally, the transport mode and pharmacology (amiloride sensitivity) of the heterologously expressed SLC9C1 and 2 or heterodimers (SLC9C1/2) and SLC9B1 must be determined.

Out of curiosity, we also performed tests ignoring the non-normal distribution of the data in Fig.4G: a one-way ANOVA gives $p = 0.014$; the post-hoc multiple comparison using the Scheffé method yields no significant difference between any of the groups. In conclusion, irrespective of the normal distribution of data, there is no significance between conditions and control in Fig. 4G.

[3] Can the authors define and explain the relationship between n_{exp} and n_{cells} . Does n_{exp} represent different days or different dishes of cells etc.? A more rigorous statistical analysis could

include batch dependence as a nested factor: this may help with the analysis of inhibitor effect.

Yes, n_{exp} represents the number of different cover slips of cells. We now specify this in the methods section. Per cover slip, the number of cells (n_{cells}) was variable (range: 1 – 10). The unequal sample size within each subgroup (also known as “unbalanced design”) complicates the implementation of a nested ANOVA: To arrive at equal samples size within each subgroup, random data can be deleted or missing data can be replaced by estimates to handle minor inequalities in sample size (see Zar JH Biostatistical analysis, Pearson, 5th edition, 2009, p. 304). More general approaches, including general linear models, can be employed to deal with this problem (Rawlings JO et al. Applied Regression Analysis. A Research Tool, Springer, 2nd edition, 1998, pp. 545ff). However, because of the wide range of different sample sizes per cover slip this approach is not feasible for our datasets.

For statistical analysis, it is preferable to have a balanced design, i.e., equal sample size per subgroup and condition. However, the performed experiments and the whole procedure including labeling, centrifuging, cell placement and imaging are complex, and it is challenging to always arrive at the same number of sperm per recording.

[4] Some statements are still lacking statistical backing, which the authors could easily calculate. For example line 494-496 appears to be based on comparison averages rather than a demonstration of significant difference among cell types in 4H data.

We changed the wording and performed the appropriate statistical analysis to determine significant differences (see reply below).

[5] In any case, a two-way ANOVA or equivalent would be more appropriate for Fig 4C, 4H, and 5H data.

A two-way ANOVA is only applicable for normally distributed data. Analogous non-parametric tests are a matter of disputes and are rather not recommended (Zar JH p.249). Here are the results concerning normal distribution: Fig.5H, normal distribution; Fig.4H, for some data *no* normal distribution; Fig.4C, Shapiro test: normal distribution, d’Agostino test one dataset ($7.7 + \text{HCO}_3^-$) *no* normal distribution; Fig. S5: Shapiro test one dataset (Alkal. Ctrl) *no* normal distribution, d’Agostino test not applicable for two data sets because n too small. Of note, in addition to the different test results, deviations from normality are relatively small in several datasets. Therefore, as the reviewer suggested, we now use two-way ANOVA for the mentioned data sets with pH and $\text{CO}_2/\text{HCO}_3^-$ (Fig. 4C), Ctrl/ACZ and cell type (Fig. 4H), or Ctrl/ACZ and alkalization/acidification (Fig. 5H and S5B). Of note, we detect that acetazolamide (ACZ) does not significantly slow down acidification following $\text{CO}_2/\text{HCO}_3^-$ perfusion in CHO cells, supporting the idea that CHO cells indeed have a very minor or almost no CA activity. The previously used KS test showed significance, probably due to the vastly different distributed data (see Fig. 4H compare control vs. ACZ in CHO cells). We now also report on the significant differences of the previously untested factors (Fig. 4C, control vs. $\text{CO}_2/\text{HCO}_3^-$; Fig. 4H, $\tau \Delta\text{pH}_i$ of cell types: sperm vs. CHO vs. HEK; Figs. 5H and S5B, $t_{1/2} \Delta\text{pH}_i$: alkalization vs. acidification). Of note, Fig. 4H illustrates a clear example of a highly significant interaction between the two factors cell type and ACZ revealed by the two-way ANOVA.

Reviewer #3 (Remarks to the Author):

Overall the authors have adequately responded to all comments except one- a final list of all proteins

identified by mass spec should be included in a Supplemental table (preferably excel) that contains details of the protein accession and the relevant quantitative information.

We have included now all Excel tables S2, S8, and S9 in the manuscript (in addition to the previous deposition in the PRIDE repository). The corresponding raw data had been already deposited to the ProteomeXchange Consortium via the PRIDE partner repository with the dataset identifier PXD036819 and we have stated that in the respective legends. Moreover, the reviewer's access code for the PRIDE repository is included. Thus full access of all data for the reviewers and the prospective readers is provided. This should remove all uncertainties about the availability of data.

REVIEWERS' COMMENTS

Reviewer #2 (Remarks to the Author):

The authors have revised their statistical approach in a thoughtful and conservative manner that satisfies my comments.

One query remains: the effect of amiloride in Fig 3E and 3I that forms the basis of one conclusion (Line 367) does not currently appear to be supported by any statistical analysis, only by two example traces. Statistics should be provided or if it is the authors contention that amiloride allows for no pH change and that statistical analysis is therefore moot, this ought ought at least to be made clear to the reader and the number of replicate observations stated.

Reviewer #2 (Remarks to the Author):

The authors have revised their statistical approach in a thoughtful and conservative manner that satisfies my comments.

One query remains: the effect of amiloride in Fig 3e and 3i that forms the basis of one conclusion (Line 367) does not currently appear to be supported by any statistical analysis, only by two example traces. Statistics should be provided or if it is the authors contention that amiloride allows for no pH change and that statistical analysis is therefore moot, this ought at least to be made clear to the reader and the number of replicate observations stated.

We now include the mean change in pH in the presence and absence of amiloride for Fig. 3e and 3i. We also provide the statistical analysis (paired test for Fig. 3e and paired Mann-Whitney-Test for Fig. 3i)) and report the significant difference between amiloride (minimal change in pH) and control condition (substantial recovery/increase in pH).